# PM$_{2.5}$ concentration assessment based on geographical and temporal weighted regression model and MCD19A2 from 2015 to 2020 in Xinjiang, China

**Weilin Quan**[1,2,3⊚], **Nan Xia**[1,2,3⊚]*, **Yitu Guo**[1], **Wenyue Hai**[1,2,3], **Jimi Song**[1,2,3], **Bowen Zhang**[1,2,3]

1 College of Geography and Remote Sensing Sciences, Xinjiang University, Urumqi, China, 2 Xinjiang Key Laboratory of Oasis Ecology, Xinjiang University, Urumqi, China, 3 Technology Innovation Center for Ecological Monitoring and Restoration of Desert-Oasis, MNR, Urumqi, China

⊚ These authors contributed equally to this work.
* xn_gis@xju.edu.cn

**Data Availability Statement:** The PM2.5 data used in this study is the hourly PM2.5 data of the national air quality real-time publishing platform of China National Environmental Monitoring Station

## Abstract

PM$_{2.5}$ is closely linked to both air quality and public health. Many studies have used models combined with remote sensing and auxiliary data to inverse a large range of PM$_{2.5}$ concentrations. However, the data's spatial resolution is limited. and better results might have been obtained if higher resolution data had been used. Therefore, this paper establishes a geographical and temporal weighted regression model (GTWR) and estimates the PM$_{2.5}$ concentration in Xinjiang from 2015 to 2020 using 1 km resolution MCD19A2 (MODIS/Terra +Aqua Land Aerosol Optical Thickness Daily L2G Global 1km SIN Grid V006) data and 9 auxiliary variables. The findings indicate that the GTWR model performs better than the simple linear regression (SLR) and geographically weighted regression (GWR) models in terms of accuracy and feasibility in retrieving PM$_{2.5}$ concentrations in Xinjiang. Simultaneously, by combining the GTWR model with MCD19A2 data, a spatial distribution map of PM$_{2.5}$ with better spatial resolution can be obtained. Next, the regional distribution of annual PM$_{2.5}$ concentrations in Xinjiang is consistent with the terrain from 2015 to 2020. The low value area is primarily found in the mountainous area with higher terrain, while the high value area is primarily in the basin with lower terrain. Overall, the southwest is high and the northeast is low. In terms of time change, the six-year PM$_{2.5}$ shows a single peak distribution with 2016 as the inflection point. Lastly, from 2015 to 2020, the seasonal average PM$_{2.5}$ concentration in Xinjiang has a significant difference, thereby showing winter (66.15µg/m$^3$)>spring (52.28µg/m$^3$)>autumn (40.51µg/m$^3$)>summer (38.63µg/m$^3$). The research shows that the combination of MCD19A2 data and GTWR model has good applicability in retrieving PM$_{2.5}$ concentration.

(http://www.cnemc.cn/) ; MODIS MCD19A2 AOD data is downloaded from Level-1 and Atmosphere Archive & Distribution System Distributed Active Archive Center (LAADS SAAC), and the download address is : https://ladsweb.modaps.eosdis.nasa.gov/search/ , the data is a multi-angle implementation of atmospheric correction algorithm , (MAIAC) product of MODIS Terra and aqua , with a resolution of 1km ; Data of temperature, relative humidity, precipitation and wind speed are from the National Earth System Science Data Center , National Science and Technology infrastructure of China (http://www.geodata.cn) . The BLH is obtained from ECMWF reanalysis data set (https://www.ecmwf.int/) .

**Funding:** This research was funded by the Xinjiang Uygur Autonomous Region university scientific research program(XJEDU2021Y011)and the "Dr. Tianchi "project(tcbs201821).The sponsors have played a role in data collection and analysis and manuscript preparation. Including the provision of laboratory hardware equipment(two portable computers), field investigation expenses in the study area, etc.

**Competing interests:** The authors have declared that no competing interests exist.

## Introduction

In addition to causing haze weather and affecting air quality, PM$_{2.5}$ (aerodynamic equivalent diameter$\leq$2.5 μm particles), a fine particulate matter in the atmosphere, will also raise the likelihood of human suffering from respiratory, cardiovascular, and cerebrovascular diseases and lung cancer [1]. The traditional ground station monitoring can obtain high-precision PM$_{2.5}$ concentration. However, the monitoring of PM$_{2.5}$ lacks spatial continuity [2]. The satellite remote sensing monitoring technology has the characteristics of high spatial resolution, wide monitoring range, and long-term all-weather real-time monitoring, which can make up for the shortage of ground monitoring stations [3]. Aerosol Optical Depth (AOD) data in satellite remote sensing are vertical integrals of aerosol extinction coefficients, which is a commonly used parameter to evaluate the degree of atmospheric pollution and climate change [4, 5]. Relevant study indicates that a significant link between AOD and PM$_{2.5}$ concentration, and AOD and other auxiliary data can be used to estimate a wide range of PM$_{2.5}$ concentration [6–8].

The MODIS (Moderate-resolution Imaging Spectroradiometer) aerosol products are the most extensively used AOD data products at present. Early PM$_{2.5}$ inversion research often used MOD04_L2 (MODIS Terra/Aqua Aerosol 5-Min L2 Swath 10km) [9, 10] and MOD04_3K (MODIS Terra/Aqua Aerosol 5-Min L2 Swath 3km) [11] data. These data have low spatial resolution. The newly released MCD19A2 product makes up for this defect effectively. This product uses the MAIAC (Multi-Angle Implementation of Atmospheric Correction) algorithm, which can provide AOD data sets with a global scale of 1 km spatial resolution [12], thereby presenting more precise data for the inversion of PM$_{2.5}$. Li et al. [13] estimated the seasonal spatial distribution of PM$_{2.5}$ in Beijing using MCD19A2 data, and the results show that MCD19A2 data have a strong ability to predict the ground PM$_{2.5}$ level on a seasonal scale basis. Gu et al. [14] determined the daily concentration of PM$_{2.5}$ in Dalian using MCD19A2 data and found that the spatial distribution of AOD and PM$_{2.5}$ tended to be consistent, and the PM$_{2.5}$ value in industrial areas was high. The above research shows that MCD19A2 is feasible in PM$_{2.5}$ inversion.

In terms of models, researchers have developed various models for retrieving PM$_{2.5}$, such as simple linear regression model (SLR) [15], machine learning model [16, 17], deep learning model [18], geographic weighted regression model (GWR) [19, 20] and GTWR [21]. Among these models, GWR models only consider the spatial correlation between PM$_{2.5}$ and other factors, ignoring the relationship between them in time. Although some machine learning models and deep learning models can consider spatiotemporal correlation, they are often modeled based on big data features, and the physical features hidden in the PM$_{2.5}$ time series and geospatial distribution are not fully utilized. Compared with the above model, GTWR considers temporal correlation and spatial characteristics comprehensively, and the model is easy to operate and achieves good accuracy even when the data set is small. Therefore, it is frequently employed in the inversion research of PM$_{2.5}$ [22]. He et al [23] used the GTWR model to reverse the 3 km resolution of China PM$_{2.5}$ spatial distribution dataset. Finding showed that the effect was superior to the D-GWR and two-stage models. Guo et al. [24] found that the GTWR model has higher spatial resolution and accuracy in estimating PM$_{2.5}$ in China than the GWR model. The aforementioned models confirm the feasibility of retrieving PM$_{2.5}$ by using GTWR model. However, the research on combining MCD19A2 data with GTWR model to express air quality quantitatively and spatially needs to be explored further.

To sum up, this paper uses MCD19A2 data, normalized difference vegetation index (NDVI), digital elevation model(DEM), and meteorological data to establish SLR, GWR, and GTWR models for each year and season in Xinjiang from 2015 to 2020 in combination with PM$_{2.5}$ ground observation data. By comparing the performance of each model, the optimal

model is selected for PM$_{2.5}$ inversion, and its spatiotemporal distribution characteristics and meteorological impact factors are analyzed to provide some reference for the monitoring and improvement of atmospheric environment quality in Xinjiang.

## Overview of the research area

Xinjiang is located in the center of the Eurasian continent, with an area of $166×10^4$ km$^2$. Influenced by complex topography and temperate continental climate, the region has scarce precipitation and serious land desertification. The Gurbantunggut and Taklamakan Deserts located in the northern and southern Tianshan Mountains have frequent sand and dust storms under the influence of high winds and serious particulate pollution [25]. In recent years, a series of aid policies such as the national western development strategy have promoted the economic development of Xinjiang, and the regional gross domestic product (GDP) of Xinjiang grew from 932.480 billion yuan to 137.978 billion yuan from 2015 to 2020, but it also brought challenges to the air quality in the region. According to the Xinjiang Ecological Environment Bulletin, only four of the 14 cities in Xinjiang will have air quality meeting the national secondary standard (35μg/m$^3$) by 2020, with 71% of the cities exceeding the pollution standard, and the annual average value of the primary pollutant PM$_{2.5}$ is 47μg/m$^3$, which is 1.3 times higher than the national secondary standard. In particular, on February 5, 2022, the cities of Wujiaqu, Shihezi, Urumqi and Changji in Xinjiang ranked among the top four worst air quality cities in the country, reaching severe pollution levels.

In summary, under the combined influence of sand and dust and human activities, the PM$_{2.5}$ pollution situation in Xinjiang is severe, and the task of timely assessment of air pollution remediation and long-term monitoring of PM$_{2.5}$ spatial and temporal distribution is imminent. The geographical distribution of environmental monitoring stations in Xinjiang are shown in (Fig 1).

## Data and method

### PM$_{2.5}$ data

The PM$_{2.5}$ data used in this study is the hourly PM$_{2.5}$ data from the national air quality real-time publishing platform of the China National Environmental Monitoring Station (http://www.cnemc.cn/). There were 35 original sites, and two monitoring sites in Kashgar region (located in southwestern Xinjiang) had more missing data and invalid values and were therefore eliminated, and the elimination principles are detailed in the cited literature [26]. Finally, 33 monitoring data sites available in Xinjiang from 2015 to 2020 are selected, and the location of the sites is shown in Fig 1. To match the AOD data in time and space, the average PM$_{2.5}$ value of each station one hour before and after the satellite transit is calculated to determine the daily, monthly, quarterly, and annual averages of each station in Xinjiang from 2015 to 2020, excluding the missing and abnormal values of PM$_{2.5}$ concentration.

### AOD data

This article has downloaded 1296 AOD data of MCD19A2 from the first, 15th, and 30th of each month in Xinjiang from 2015 to 2020. The data are obtained from National Aeronautics and Space Administration (NASA) with a 1 km resolution and the data band used in the study is 550 nm [27]. Moreover, the accuracy of this data has been verified by many scholars, such as Bilal et al [28] and Filonchyk et al [29]. The batch processing of QA value screening, reprojection and mosaic is completed through IDL code programming, and the AOD daily mean data of sinusoidal projection are obtained. The reproject raster tool is used in ENVI 5.3 to reproject

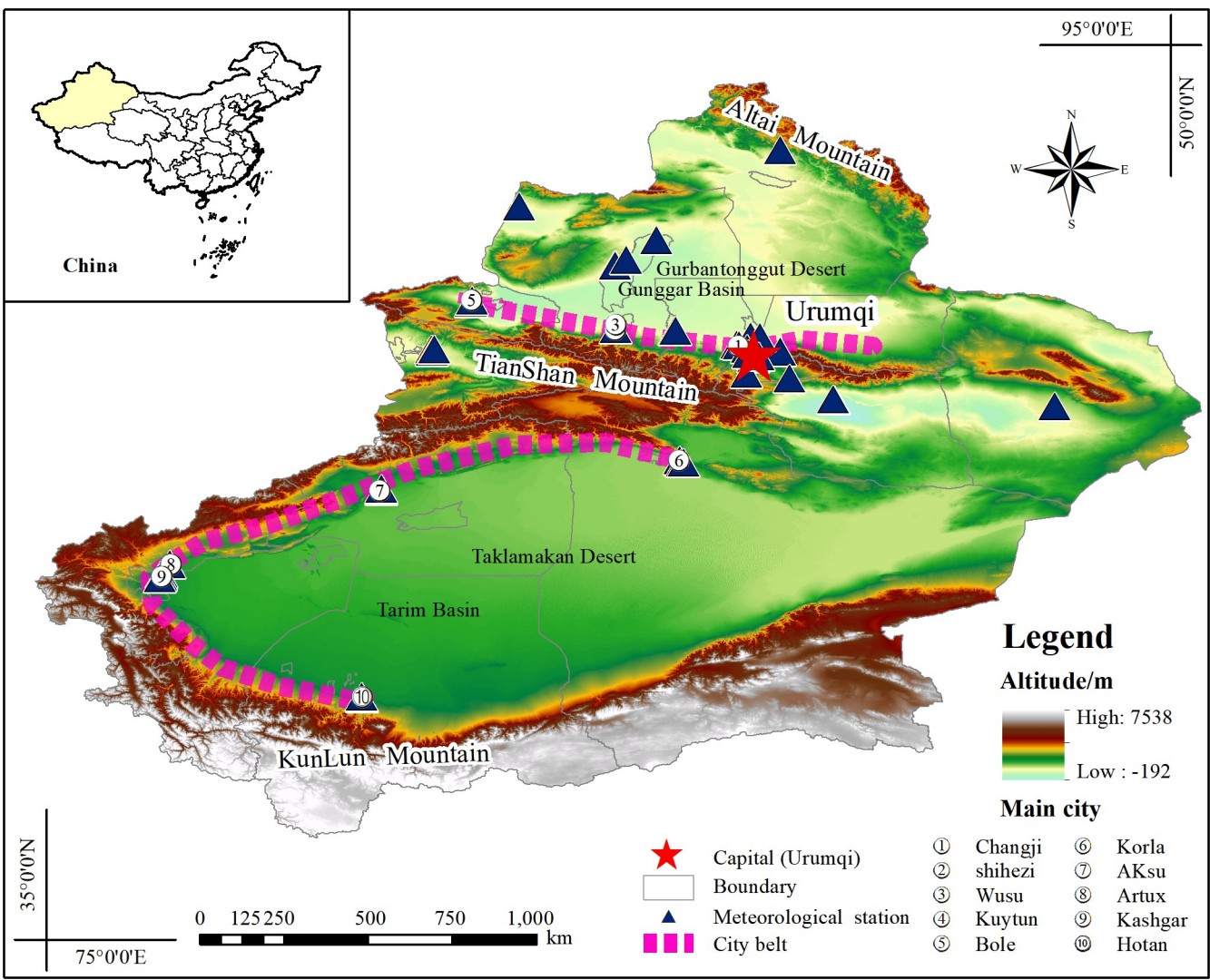

**Fig 1. Location of the research area and the placement of PM2.5 monitoring stations.**

each image to WGS84, the resize data tool is used to resample to 1 km, and the IDL code is applied to calculate the average value at the beginning, middle, and end of each month to replace the monthly average value. After this process, the AOD data still have missing values in some areas. This study established 0.01˚×0.01˚ fishing net in ArcGIS 10.5 to fill the missing value, and the monthly AOD value is extracted to the corresponding point in the fishing net. After eliminating the null value point, the ordinary Kriging method in the geostatistical analysis module is used for interpolation (Fig 2). The data used in the interpolation process are normally distributed after log transformation. The standard root mean square of the interpolation results is greater than 0.80, the standard average is between 0.03 and 0.06, and the average standard error and root mean square error are close to 0, meeting the accuracy requirements. The statistical results of interpolation indicators for each year from 2015 to 2020 are shown in Table 1. At present, relevant studies have shown that the AOD data interpolated by the ordinary Kriging method have high accuracy. For example, Zhao Fei [30] compared the AOD interpolation in Beijing Tianjin Hebei region with the inverse distance weighting method, the

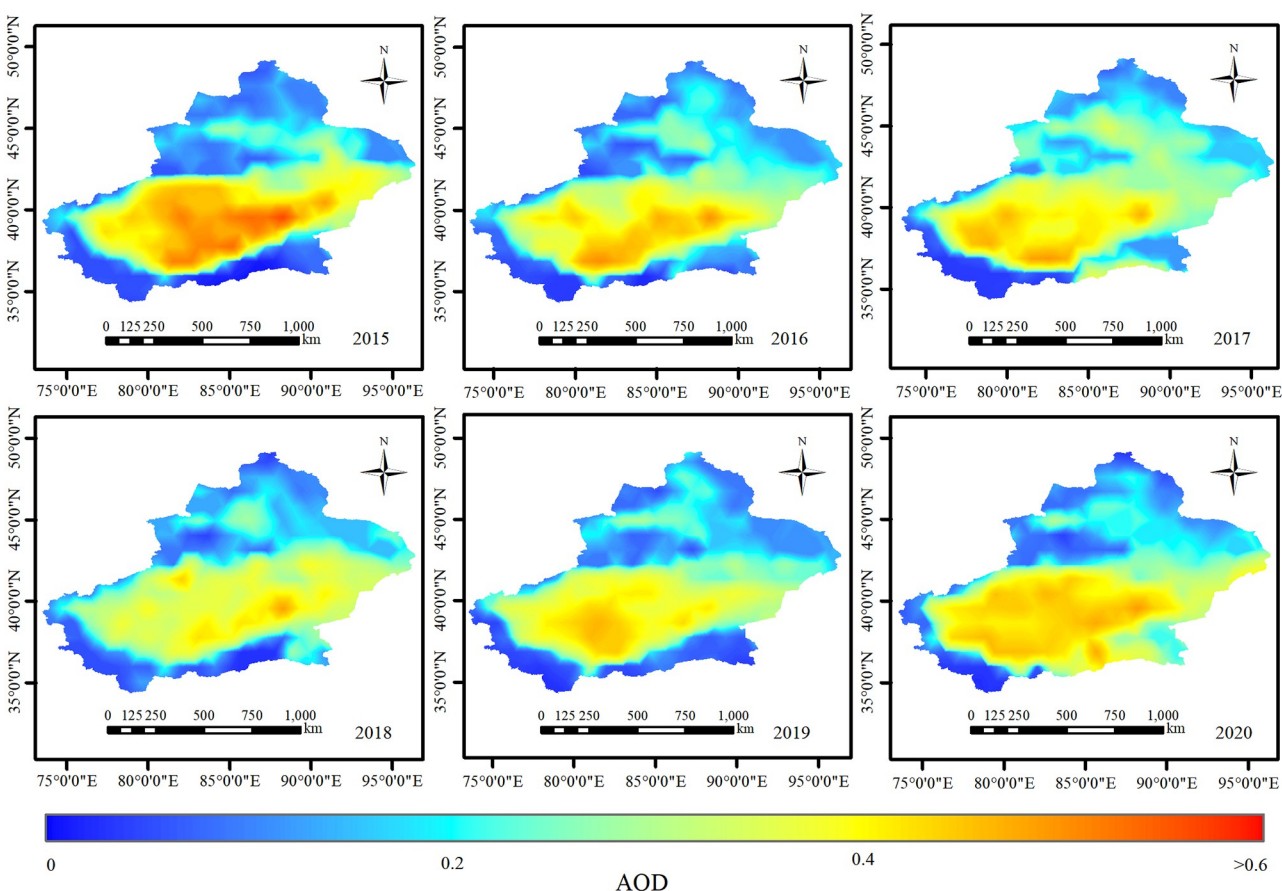

**Fig 2. AOD interpolation diagram from 2015–2020.**

ordinary Kriging method, the natural neighborhood method, and the spline function method. He found that the ordinary Kriging method has high interpolation accuracy. Baokei LV et al. [31] used the ordinary Kriging interpolation method to get the AOD data covering the Beijing Tianjin Hebei region for 365 days, and the data accuracy is high.

## Auxiliary data

The monthly temperature (TEM), precipitation (PRE), relative humidity (RH), wind speed (WS), normalized difference vegetation index (NDVI) and evapotranspiration (ET) data from 2015 to 2020 are gathered with a 1 km resolution from the National Earth System Science Data Center, National Science and Technology infrastructure of China. The atmospheric boundary

**Table 1. Indicator statistics of Kriging interpolation results in 2015–2020.**

| Index | Kriging interpolation index results in 2015–2020 | | | | | |
|---|---|---|---|---|---|---|
| | 2015 | 2016 | 2017 | 2018 | 2019 | 2020 |
| Mean Standardized | 0.03 | 0.06 | 0.05 | 0.04 | 0.03 | 0.04 |
| Average Standard Error | 0.09 | 0.12 | 0.08 | 0.09 | 0.09 | 0.09 |
| Root-Mean-Square | 0.07 | 0.07 | 0.05 | 0.05 | 0.08 | 0.06 |
| Root-Mean-Aquare Standardized | 0.94 | 0.85 | 0.80 | 0.84 | 0.96 | 0.81 |

layer height (BLH) is obtained from European Centre for Medium-Range Weather Forecasts (ECMWF, https://www.ecmwf.int/) reanalysis data set, resampled to 1 km. The digital elevation model (DEM) data was gathered from the Resource and Environment Science and Data Center (https://www.resdc.cn/). All data has a resolution of 1 km to match the time and space, and the corresponding AOD and meteorological data values are extracted from the meteorological monitoring stations for subsequent modeling.

## Method

### Humidity correction and vertical correction

The AOD value obtained by the MODIS satellite represents the vertical integration of the aerosol extinction coefficient, whereas the PM$_{2.5}$ concentration data monitored by the station represents the concentration of particles near the ground. There is a large spatial difference between the two, so it is necessary to use BLH data to vertically correct the AOD [32, 33]. The formula is:

$$R_{AOD} = \frac{AOD}{BLH} \qquad (1)$$

Considering that some particles in the atmosphere will expand after absorbing water vapor, thus affecting the aerosol extinction coefficient, RH data is used for humidity correction of AOD [34]. The formula is:

$$R_{AOD} = \frac{AOD}{f(RH)} \qquad (2)$$

Wherein, $f(RH)$ represents the humidity influence factor.

### Multiple collinearity diagnosis

Before modeling, nine independent variables of the input model, namely, RH, WS, PRE, TEM, BLH, NDVI, ET, DEM and AOD data, are collinear diagnosed in SPSS Statistics 17.0 to avoid invalid model estimates or low prediction accuracy due to high correlation between independent variables. Variance expansion factor (VIF) [35] is the ratio of variances that are multicollinear among explanatory variables to variances that are not. It is used to characterize the degree of collinearity among variables. The formula is:

$$VIF = \frac{1}{1 - R^2} \qquad (3)$$

### Simple linear regression (SLR)

The relationship between two variables with correlation can be described using a linear model, as follows:

$$y = A + Bx + \varepsilon \qquad (4)$$

where A and B are constants, and $\varepsilon$ is a random variable. Simple linear regression model can capture the most fundamental binary association between AOD and PM$_{2.5}$. [36].

### Geographically weighted regression (GWR)

GWR was first proposed by Fotheringham et al. [37]; it is an improved SLR model, which is an extension of the global regression model to solve the problem of spatial data nonstationarity.

Compared with the SLR model, GWR model can consider the effect of many factors on the spatial relationship between AOD and PM$_{2.5}$. The formula is:

$$Y_j = \beta_0\left(u_j, v_j\right) + \sum_k \beta_k\left(u_j, v_j\right)X_{jk} + \varepsilon_j \tag{5}$$

Where $(u_j, v_j)$ represents the coordinates of the point j in space; $\beta_0(u_j, v_j)$ indicates intercept; $\beta_k(u_j, v_j)$ represents the regression coefficient; $X_{jk}$(k = 1,2,. . ., P) represents the independent variable value at j; $\varepsilon_j$ is the residual.

### Spatiotemporal geographically weighted regression (GTWR)

GTWR model is a further extension of GWR model. Huang et al. [38] introduced the time effect based on the GWR model and developed the GTWR model to deal with the non-stationary problems in time and space. GTWR can be written as follows:

$$Y_i = \beta_0\left(u_j, v_j, t_j\right) + \sum_k \beta_k\left(u_j, v_j, t_j\right)X_{jk} + \varepsilon_j \tag{6}$$

where $(u_j, v_j, t_j)$ represents the space–time coordinates of j in the space point; $\beta_k(u_j, v_j, t_j)$ is the regression coefficient; $\varepsilon_j$ is the residual.

The flowchart of main method is sown in Fig 3.

## Results and discussion

### Humidity correction and vertical correction

According to the statistics on the association between AOD and PM$_{2.5}$ concentrations in Xinjiang from 2015 to 2020, the overall correlation coefficient between AOD and PM$_{2.5}$ concentrations in six years is 0.193, and the correlation coefficient in each year is between -0.083 and 0.153, with poor direct correlation. The revised results show that the correlation coefficient between AOD and PM$_{2.5}$ concentration in each year is significantly improved compared with that prior to correction, and the vertical correction result has been improved to 0.579, which is superior to humidity correction (0.363). Therefore, the AOD data after vertical correction is selected for subsequent remote sensing inversion. Table 2 shows Pearson's r for each year and the whole of 2015–2020 under the confidence level of 0.01.

### Multiple collinearity diagnosis

The value of VIF is greater than 1. When the VIF value is closer to 1, no multicollinearity exists. When VIF>10, strong multicollinearity is considered. The annual and seasonal independent variable VIF values after multicollinearity diagnosis are displayed in Table 3. The findings reveal that the VIF values of each factor are between 1.17 and 8.10 after removing the variables with strong correlation, and there is no serious multicollinearity, which can be used for subsequent modeling.

### Model comparison

The diagnosis results above show that the multicollinearity between independent variables is weak and has the conditions to build a model. Therefore, using the processed PM$_{2.5}$ data and other auxiliary variable data of Xinjiang from 2015 to 2020, the GWR and GTWR models of each year and season in Xinjiang from 2015 to 2020 are established in ArcGIS10.5 through the GTWR plug-in developed by Huang et al. [38], and the SLR model was established in Excel 2016. Fig 4 shows the scatter plot of the estimated and measured PM$_{2.5}$ concentrations for

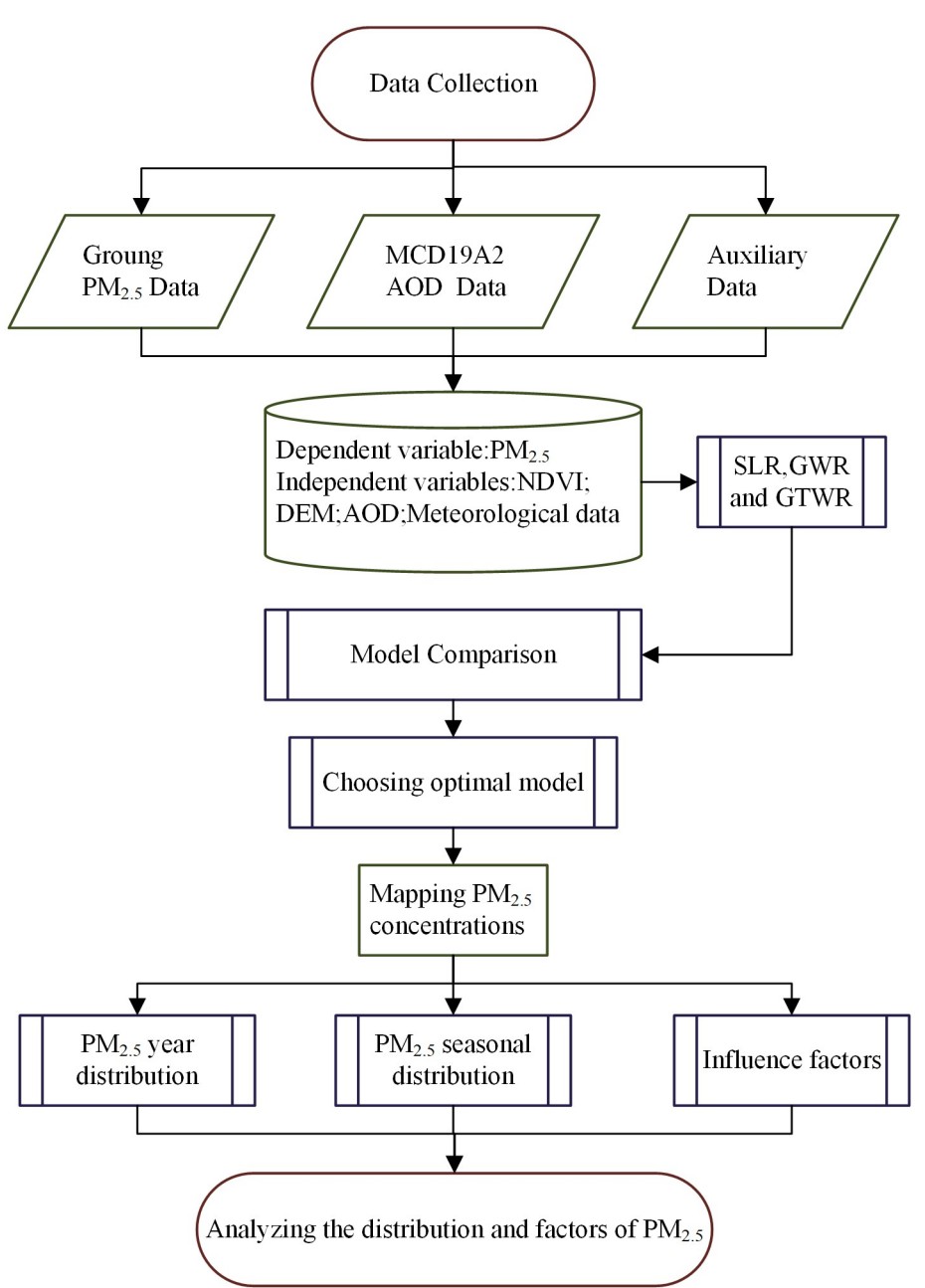

**Fig 3. Workflow outline.**

**Table 2. Correlation table of AOD–PM$_{2.5}$ concentration under different correction states.**

| Revision status | AOD-PM$_{2.5}$ Pearson's r | | | | | |
|---|---|---|---|---|---|---|
| | **2015** | **2016** | **2017** | **2018** | **2019** | **2020** |
| No revision | 0.085 | -0.083 | 0.005 | 0.012 | 0.036 | 0.153 |
| Humidity correction | 0.328 | 0.299 | 0.375 | 0.374 | 0.364 | 0.397 |
| Vertical correction | 0.472 | 0.485 | 0.512 | 0.518 | 0.543 | 0.580 |

**Table 3. Multicollinearity diagnostic tables for years and seasons, 2015–2020.**

| Time | VIF | | | | | | | | | |
|---|---|---|---|---|---|---|---|---|---|---|
| | Number | RH | WS | PRE | TEM | BLH | NDVI | ET | DEM | AOD |
| 2015 | 8 | 3.10 | 1.50 | 2.61 | 7.72 | 5.24 | 2.66 | - | 1.53 | 1.56 |
| 2016 | 8 | 2.89 | 1.63 | 2.63 | 7.68 | 3.94 | 3.59 | - | 1.43 | 1.78 |
| 2017 | 8 | 2.23 | 1.58 | 2.66 | - | 6.40 | 2.79 | 6.94 | 1.47 | 2.41 |
| 2018 | 8 | 2.85 | 1.43 | 2.56 | 8.10 | 4.22 | 2.78 | - | 1.42 | 1.68 |
| 2019 | 8 | 2.61 | 1.44 | 2.53 | 7.83 | 4.32 | 2.38 | - | 1.54 | 1.99 |
| 2020 | 8 | 2.32 | 1.44 | 2.61 | - | 7.91 | 2.93 | 7.14 | 1.34 | 1.80 |
| Spring | 8 | 2.04 | 1.40 | 2.05 | 5.13 | 3.99 | 2.05 | - | 2.52 | 1.17 |
| Summer | 9 | 2.37 | 1.27 | 2.55 | 2.90 | 1.54 | 1.43 | 1.26 | 2.14 | 1.18 |
| Autumn | 8 | 1.90 | 1.33 | 2.12 | - | 4.55 | 1.56 | 4.29 | 1.35 | 1.62 |
| Winter | 9 | 3.10 | 1.23 | 1.65 | 2.91 | 3.84 | 1.39 | 4.31 | 1.39 | 1.83 |

SLR, GWR, and GTWR models for 2015–2020. Table 4 summarizes the evaluation index of each model. In this study, determination coefficient ($R^2$), root mean square error (RMSE), mean absolute error (MAE) and Akaike information criterion (AICc is a measure of model performance, that helps to compare different regression models, and models with lower AICc values have better performance), were used to compare the overall performance of the GTWR, GWR, and SLR models.

According to Fig 4 and Table 4, the fitting effect of the SLR model is the worst, and its $R^2$ in each year does not exceed 0.3. The RMSEs of the SLR model in 2017 and 2018 are the largest, reaching 55.62μg/m$^3$ and 51.48μg/m$^3$, respectively. This result shows that the estimated value of the SLR model does not have a good fitting relationship with the measured value. The GWR model $R^2$ is between 0.66 and 0.76, and the six-year average $R^2$ is 0.64. Compared with the SLR model, the average RMSE and MAE are reduced by 25.27μg/m$^3$ and 16.30μg/m$^3$, respectively. Its estimation effect is significantly improved compared with the SLR model, but a gap still exists when compared with the GTWR model. From 2015 to 2020, the fitting $R^2$ of the GTWR model in each year was further improved on the basis of GWR. The six-year $R^2$ was 0.69, 0.72, 0.77 0.70, 0.73, and 0.70, with an average $R^2$ of 0.72, which is greater than those for the SLR and GWR models. The average RMSE and MAE decreased by 25.77μg/m$^3$ and 16.59μg/m$^3$ and 0.40μg/m$^3$ and 0.29μg/m$^3$ on the basis of SLR and GWR models, respectively. In 2017, the best year for GTWR model fitting was achieved, wherein $R^2$ = 0.77, RMSE = 23.02 μg/m$^3$, and AICc = 3732.62. Therefore, the GTWR model has more reliable estimation effect and smaller prediction error when compared to the SLR and GWR models. In general, under the synergetic effect of data time and space dimensions, GTWR shows a higher PM$_{2.5}$ estimation performance. Its estimated PM$_{2.5}$ value is in good consistency with the measured value and outperforms the SLR and GWR models in terms of stability and accuracy.

Fig 5 shows the estimation effects of SLR, GWR, and GTWR models on PM$_{2.5}$ concentration in spring (March–May), summer (June–August), autumn (September–November), and winter (December–February) from 2015 to 2020. According to Fig 5 and Table 4, the estimation effect of the GTWR model on seasonal PM$_{2.5}$ in Xinjiang is still superior to that of the SLR and GWR models. The data modeled here are six-year averages for each season from 2015–2020. The GTWR model has the best estimation effect in summer, where the error is the smallest. Its $R^2$ is 0.83, RMSE = 7.72μg/m$^3$, and MAE = 5.62μg/m$^3$. Thus, its estimated PM$_{2.5}$ value can reflect the actual PM$_{2.5}$ value well. The $R^2$ in spring and autumn are 0.68 and 0.78, respectively, with good estimation effect. The fitting $R^2$ in winter is the lowest, and RMSE and MAE, which are 31.31μg/m$^3$ and 22.13μg/m$^3$, respectively, are the largest. Additionally, the

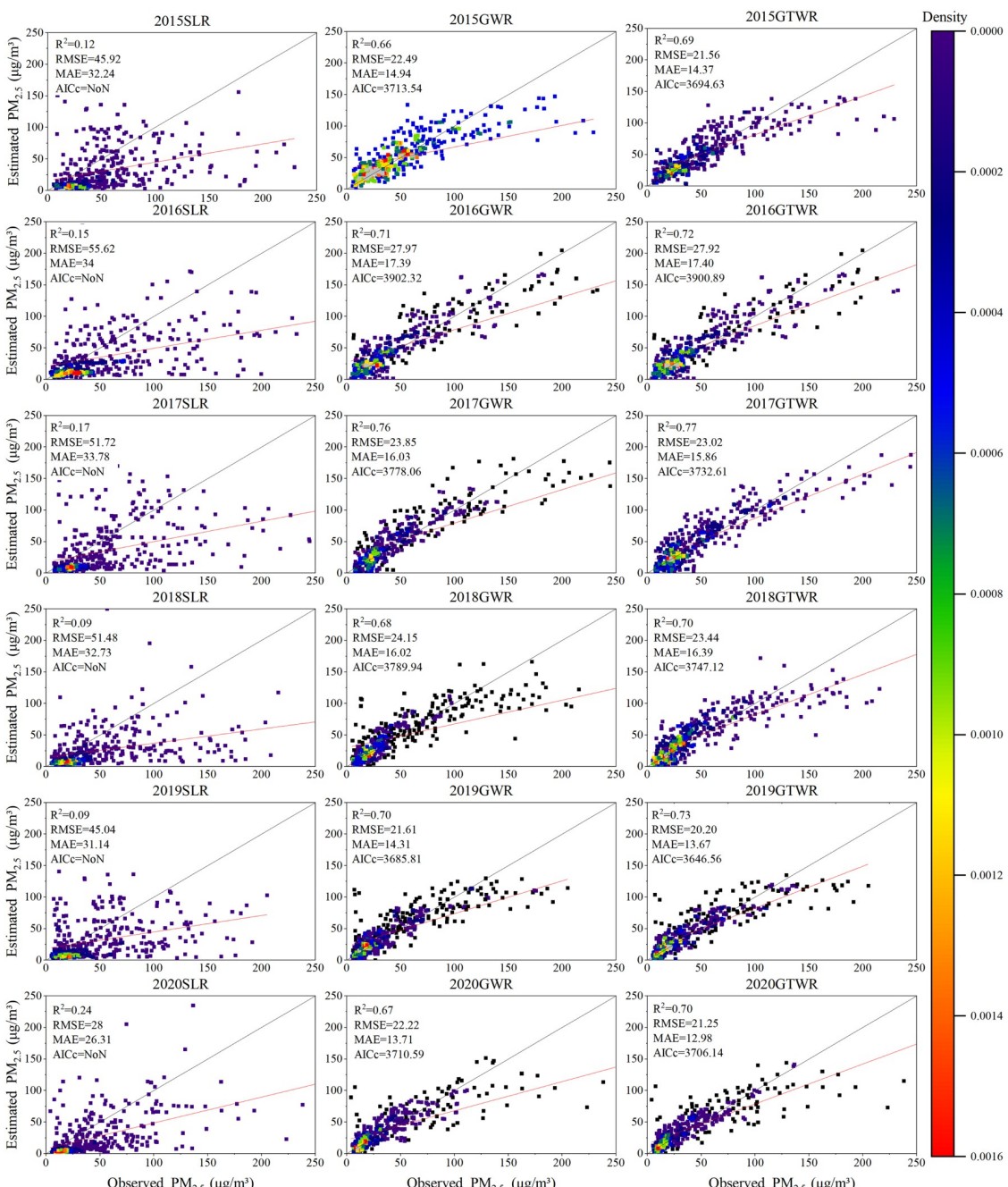

**Fig 4. Scatter density map of fitting PM$_{2.5}$ values estimated by SLR, GWR and GTWR models and PM$_{2.5}$ values monitored by stations from 2015 to 2020.**

AICc value of the GTWR model in each season decreased by 13.48–214.10 compared with that of the GWR model, indicating that the estimated probability distribution of the GTWR model is closer to the true distribution. Therefore, compared with the SLR and GWR models, the GTWR model that combines AOD, meteorological data, topographic data, and other auxiliary data can capture the temporal and spatial changes of PM$_{2.5}$ to a large extent. Therefore, it is more suitable for predicting PM$_{2.5}$ concentrations in various years and seasons in Xinjiang.

**Table 4. Average model evaluation index of each year and four seasons from 2015 to 2020.**

| Year | Model | $R^2$ | RMSE | MAE | AICc |
|---|---|---|---|---|---|
| 2015 | SLR | 0.12 | 45.92 | 32.24 | - |
| | GWR | 0.66 | 22.49 | 14.94 | 3713.54 |
| | GTWR | 0.69 | 21.56 | 14.37 | 3694.63 |
| 2016 | SLR | 0.15 | 55.62 | 34 | - |
| | GWR | 0.71 | 27.97 | 17.38 | 3902.32 |
| | GTWR | 072 | 27.92 | 17.40 | 3900.89 |
| 2017 | SLR | 0.17 | 51.72 | 33.78 | - |
| | GWR | 0.76 | 23.85 | 16.03 | 3778.06 |
| | GTWR | 0.77 | 23.02 | 15.86 | 3732.62 |
| 2018 | SLR | 0.09 | 51.48 | 32.73 | - |
| | GWR | 0.68 | 24.15 | 16.02 | 3789.94 |
| | GTWR | 0.70 | 23.44 | 16.39 | 3747.12 |
| 2019 | SLR | 0.09 | 45.04 | 31.14 | - |
| | GWR | 0.70 | 21.61 | 14.31 | 3685.81 |
| | GTWR | 0.73 | 20.20 | 13.67 | 3646.56 |
| 2020 | SLR | 0.24 | 40.28 | 26.31 | - |
| | GWR | 0.67 | 22.22 | 13.71 | 3710.59 |
| | GTWR | 0.70 | 21.25 | 12.97 | 3706.14 |
| Spring | SLR | 0.07 | 51.59 | 30.3 | - |
| | GWR | 0.66 | 25.88 | 15.64 | 5658.61 |
| | GTWR | 0.68 | 24.97 | 15.07 | 5645.13 |
| Summer | SLR | 0.03 | 23.86 | 14.75 | - |
| | GWR | 0.65 | 10.92 | 7.55 | 4613.85 |
| | GTWR | 0.83 | 7.72 | 5.62 | 4399.75 |
| Autumn | SLR | 0.16 | 28.42 | 20.48 | - |
| | GWR | 0.65 | 14.11 | 9.72 | 4958.73 |
| | GTWR | 0.78 | 11.39 | 7.85 | 4852.31 |
| Winter | SLR | 0.059 | 73.33 | 55.55 | - |
| | GWR | 0.49 | 37.19 | 27.43 | 6141.30 |
| | GTWR | 0.64 | 31.31 | 22.13 | 6119.09 |

In summary, the GTWR model can simultaneously consider the effects of spatiotemporal distance on each variable and can solve the problem of spatiotemporal instability in the inversion. When using the same data to estimate Xinjiang PM$_{2.5}$, the GTWR model that considers the temporal and spatial variations of AOD-PM$_{2.5}$ has higher estimation accuracy and performance than SLR and GWR models, with the six-year model fitting $R^2$ range of 0.69–0.76 and the four-season model fitting $R^2$ range of 0.64–0.83. In addition, we also compared the GTWR model with other models and found that the GTWR model has comparable or even better performance with some deep learning models and machine learning models. For example, Ni et al., [39] used the BPNN model combined with AOD data and other meteorological factors to estimate PM$_{2.5}$ concentrations in the Beijing-Tianjin-Hebei region of China, with a model $R^2$ of 0.68; Nabavi et al. [40] studied the performance of four machine learning algorithms to estimate PM$_{2.5}$ concentrations in the Tehran region, and the results showed that random forest (RF) performed the best with $R^2$ = 0.68. GTWR model also outperformed other statistical and combinatorial models, such as the artificial neural network (ANN) model ($R^2$ = 0.73) used by

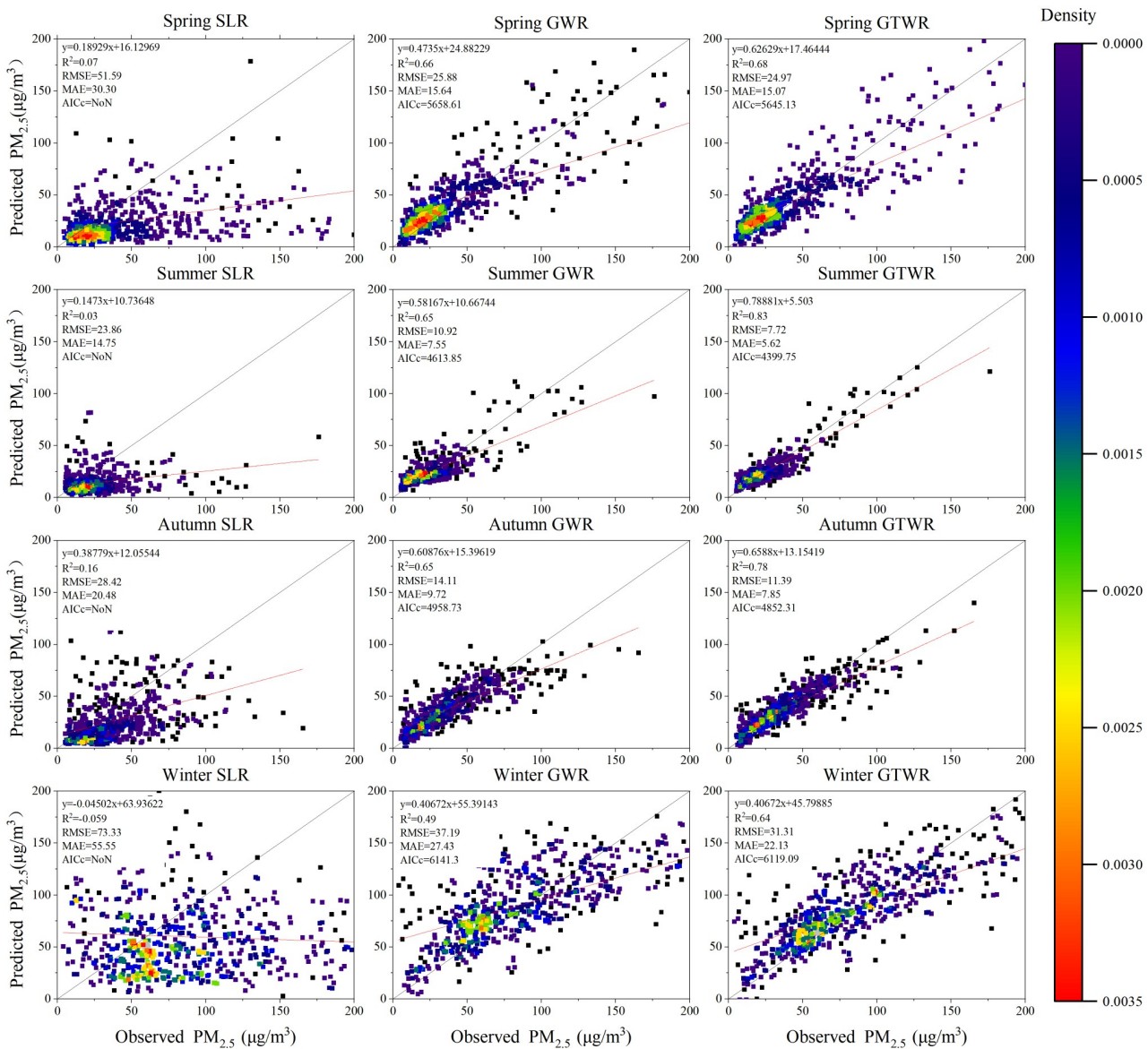

**Fig 5. Scatter density map of fitting PM$_{2.5}$ values estimated by SLR, GWR and GTWR models and PM$_{2.5}$ values monitored by stations in the four seasons of 2015 to 2020.**

Hao et al. [41] to estimate PM$_{2.5}$ concentrations in Beijing from 2008 to 2013, as well as the ordinary least squares (OLS) model ($R^2$ = 0.63), linear regression model [42] ($R^2$ range 0.59–0.73) and combined model of random forest model and mixed effects model [43] ($R^2$ = 0.71), etc. Therefore, the GTWR model can better explain the spatial and temporal distribution of PM$_{2.5}$ in Xinjiang. Traditional simple linear regression models, generalized weighted models, and two-stage models can also be used to invert PM$_{2.5}$, but their accuracy of PM$_{2.5}$ estimation is limited, whereas machine learning models and deep learning models can express the nonlinear AOD-PM$_{2.5}$ relationship but frequently fail to account for the physical mechanism and spatiotemporal heterogeneity of the AOD-PM$_{2.5}$ relationship.

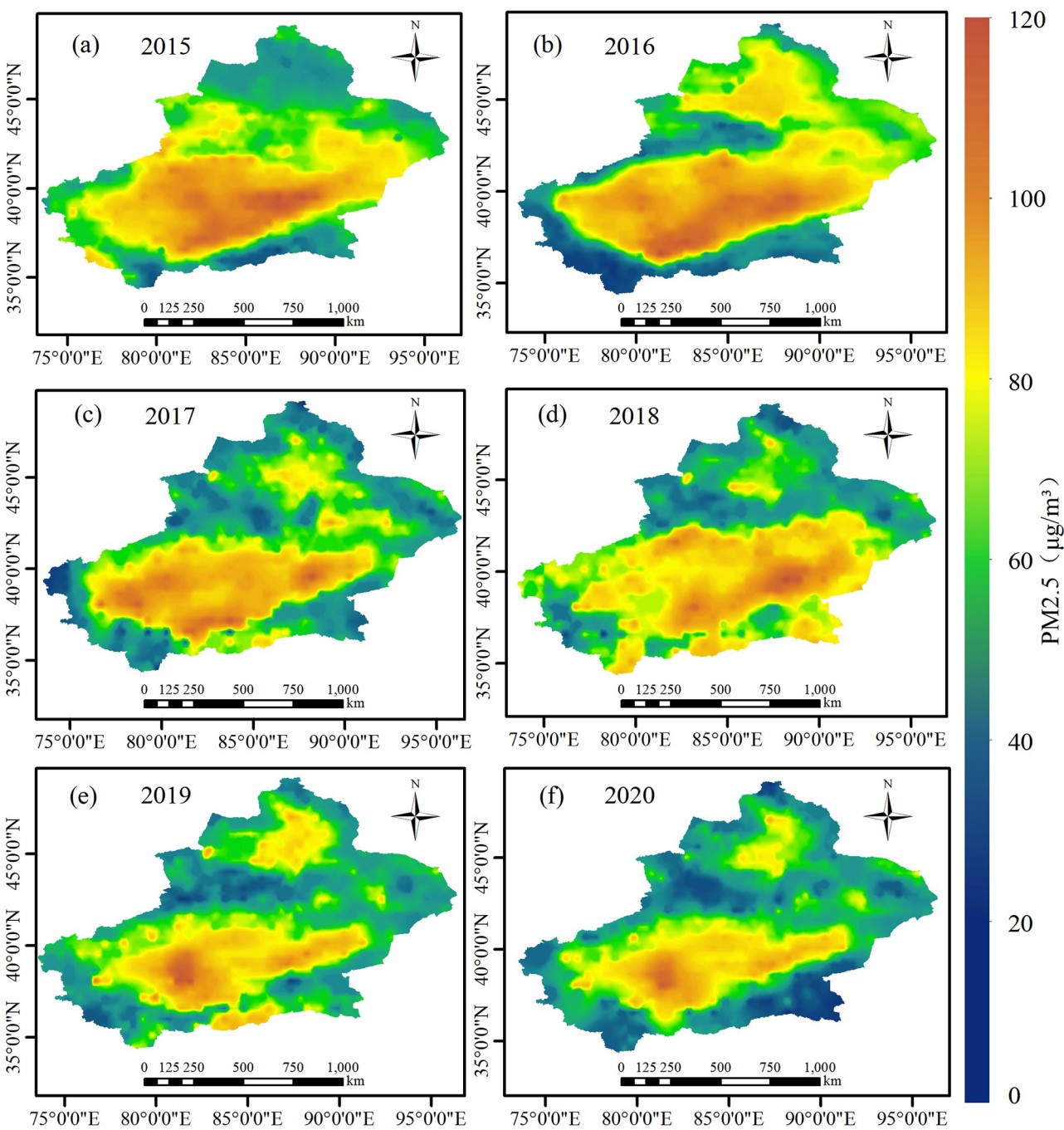

**Fig 6. Spatial distribution map of PM2.5 concentration in Xinjiang from 2015 to 2020 inverted by GTWR model.**

## PM2.5 year distribution in Xinjiang

The findings of the above model comparison reveal that the GTWR model is more accurate in estimating the PM2.5 concentration in Xinjiang. Therefore, IDW method is used in Arc-GIS10.5 to estimate the PM2.5 concentration in Xinjiang from 2015 to 2020 (Fig 6). In general, the annual distribution of PM2.5 in Xinjiang is very consistent with the topography. High

altitude areas are less affected by human activities and have more precipitation, which has a removal effect on pollutants, so the PM$_{2.5}$ concentration value is lower. Low altitude areas have high intensity of human activities and more PM$_{2.5}$ emissions, in addition, the terrain of low altitude areas is surrounded by mountains on all sides, which hinders the removal and diffusion of PM$_{2.5}$, resulting in higher pollution levels of PM$_{2.5}$, such as the Junggar Basin in the north and the Tarim Basin in the south. From 2015 to 2020, under the implementation of a series of pollution prevention policies such as 《The Action Plan for the Prevention and Control of Air Pollution》 and 《The Three-Year Action Plan for Winning the Blue Sky Defense War》, the total PM$_{2.5}$ level in Xinjiang was decreasing, and the area of high value areas of PM$_{2.5}$ continued to decrease, especially in the Tarim Basin. In addition, taking Tianshan Mountains as the boundary, the PM$_{2.5}$ concentrations in the south and north of the Tianshan Mountains are obviously different, thereby showing a geographical distribution pattern of high in the southwest and low in the northeast, which is compatible with the research results of Fu et al. [44] on the inversion of PM$_{2.5}$ in Xinjiang.

To investigate the regional peculiarities of PM$_{2.5}$ dispersion in Xinjiang further, the six-year average PM$_{2.5}$ value from 2015 to 2020 was calculated (Fig 7). It also analyzes the PM$_{2.5}$ hot spots of the urban agglomeration on the northern slope of the Tianshan Mountain, the urban agglomeration on the northern slope of Kunlun Mountain and Turpan City in the east of Xinjiang. The urban agglomerations on the north slope of Tianshan Mountains, mainly including Yining, Bole, Karamay, Urumqi, Shihezi, and Changji, are located in the Junggar Basin in the north of Xinjiang. Their six-year average PM$_{2.5}$ is 36.63μg/m$^3$, 41.45μg/m$^3$, 49.69μg/m$^3$, 49.63μg/m$^3$, 58.17μg/m$^3$, and 52.44μg/m$^3$. The PM$_{2.5}$ pollution in these areas is mainly related to dust weather, industrial emissions in economic activities, and fossil fuel emissions during heating period [45]. The urban agglomerations on the northern slope of the Kunlun Mountains, including Korla, Aksu, Artux, Hotan, and Kashgar, are located at the edge of the Tarim Basin in the south of Xinjiang. Their six-year average PM$_{2.5}$ values are 67.26μg/m$^3$, 71.08μg/m$^3$, 48.45μg/m$^3$, 73.06μg/m$^3$ and 61.41μg/m$^3$ respectively, which are overall high compared with the cities on the northern slope of Tianshan Mountain. The Taklimakan Desert, located in the Tarim Basin, is the world's second biggest movable desert, providing a vast number of dust sources for northwest China. [46]. Therefore, under the influence of high temperatures, low precipitation, and frequent sandstorms, the PM$_{2.5}$ level in the Tarim Basin becomes extremely high [47, 48]. Wind-blown sand erosion in the Tarim Basin is also an important contributor to PM$_{2.5}$ [49]. The annual average number of sandstorm days in various regions of southern Xinjiang is 0–32 d. Hotan and Korla are areas with high incidence of sandstorms [50]. Therefore, PM$_{2.5}$ levels in these regions are often higher than those in other regions. Turpan in the east of Xinjiang is also a hot spot of PM$_{2.5}$, with an average PM$_{2.5}$ concentration of 57.04μg/m$^3$ in six years. The area is a basin surrounded by mountains, thereby hindering the diffusion of particles. Thus, the PM$_{2.5}$ concentration is relatively high.

Fig 8 shows that the yearly average concentration change trend of PM$_{2.5}$ in Xinjiang from 2015 to 2020 takes 2016 as the turning point, showing a single peak distribution. Before 2016, the concentration of PM$_{2.5}$ in Xinjiang showed an increasing trend, from 64.7μg/m$^3$ in 2015 to 77.8μg/m$^3$ in 2016. After 2016, the concentration of PM$_{2.5}$ in Xinjiang has decreased year by year, with an average annual decrease of about 10.85μg/m$^3$. The statistics of 《The bulletin on the ecological environment of Xinjiang Uygur Autonomous Region》 from 2015 to 2020 indicates that with the formulation of the 《2016 Key Points for the Prevention and Control of Air pollution in Xinjiang Uygur Autonomous Region》 by the people's Government of Xinjiang Uygur Autonomous Region in 2016 and the implementation of the 《Action Plan for the Prevention and Control of Unmanned Air in Xinjiang Autonomous Region》 in 2016, the pollution prevention and control work has been continuously implemented, and the environmental

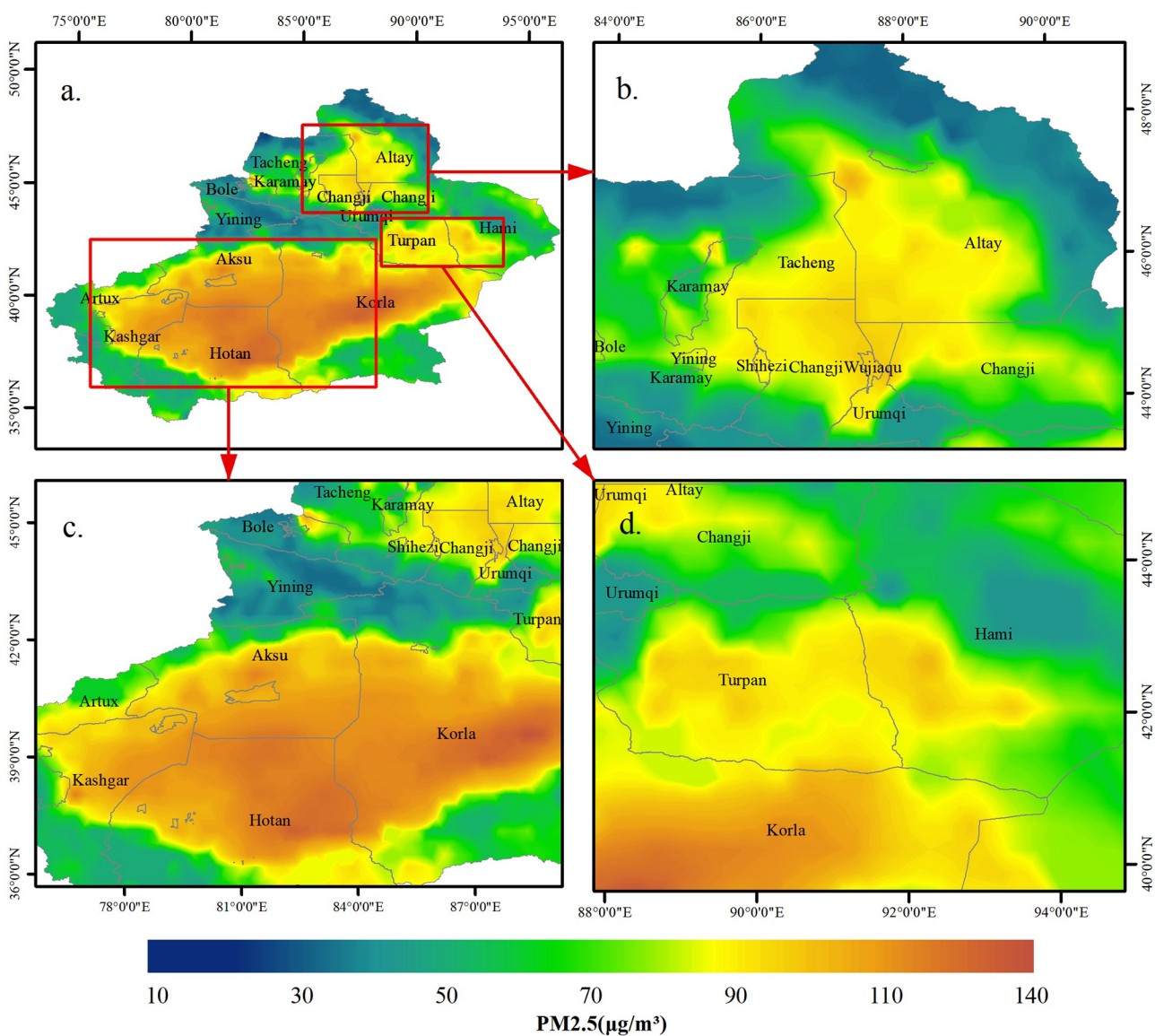

**Fig 7.** Spatial distribution map of six year average $PM_{2.5}$ concentration in Xinjiang and local map of major hotspots: a. Spatial distribution map of six year average $PM_{2.5}$, b. Urban agglomeration on the North Slope of Tianshan Mountains, c. Urban agglomeration on the North Slope of Kunlun Mountains, d. Turpan.

situation in Xinjiang has improved. The concentration of inhalable particles (Fig 9a) increased from 141 in 2016μg/m³ to μg/m³. The number of days with excellent ambient air quality increased from 67% to 75.6% (Fig 9b). In 2015–2018, the number of local sandstorms (Fig 9c) decreased from 27 days to 19 days, so the concentration of $PM_{2.5}$ also decreased.

**Seasonal distribution of PM2.5 in Xinjiang.** The spatial distribution of $PM_{2.5}$ concentration in spring, summer, autumn, and winter of 2015–2020 in Xinjiang is shown in Fig 10. The high $PM_{2.5}$ level areas in the summers of 2015–2020 are located in the Tarim Basin, whereas the high $PM_{2.5}$ concentration areas in spring, autumn, and winter are distributed in the main human activity areas along the southern and northern Xinjiang, including Shihezi, Changji, and Urumqi in the northern Xinjiang, as well as Korla, Aksu, Artux, Kashgar, and Hotan

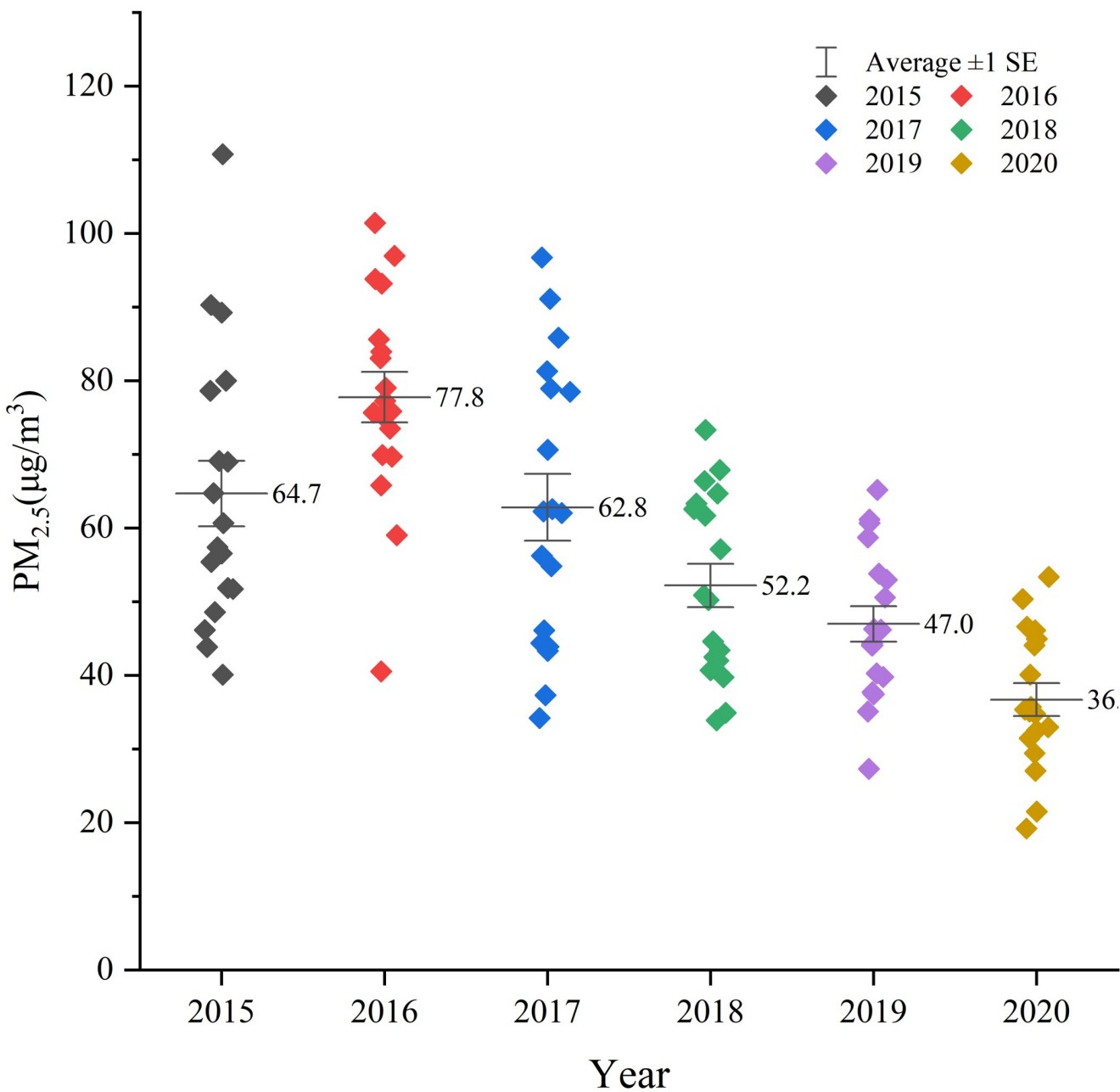

**Fig 8. Annual average PM$_{2.5}$ concentration change from 2015 to 2020.**

regions. Their average PM$_{2.5}$ concentrations are 65.56µg/m$^3$, 58.37µg/m$^3$, 65.77µg/m$^3$ and 74.46µg/m$^3$, 79.04µg/m$^3$, 56.10µg/m$^3$, 74.18µg/m$^3$, 81.68µg/m$^3$. According to the latest ambient air quality standard, the average PM$_{2.5}$ Class I standard concentration level is 15 µg/m$^3$, and the PM$_{2.5}$ secondary standard limit is 35µg/m$^3$ [51]. Obviously, the PM$_{2.5}$ levels in the aforementioned cities exceed the national secondary standard, and the PM$_{2.5}$ levels in the southern Xinjiang cities is generally higher than that in the northern Xinjiang. The PM$_{2.5}$ value in Hotan area is the highest, exceeding the limit of Level II standard by 133.37% and those in Korla, Aksu, and Artux by 112.74%, 125.83%, and 111.94%, respectively. Frequent

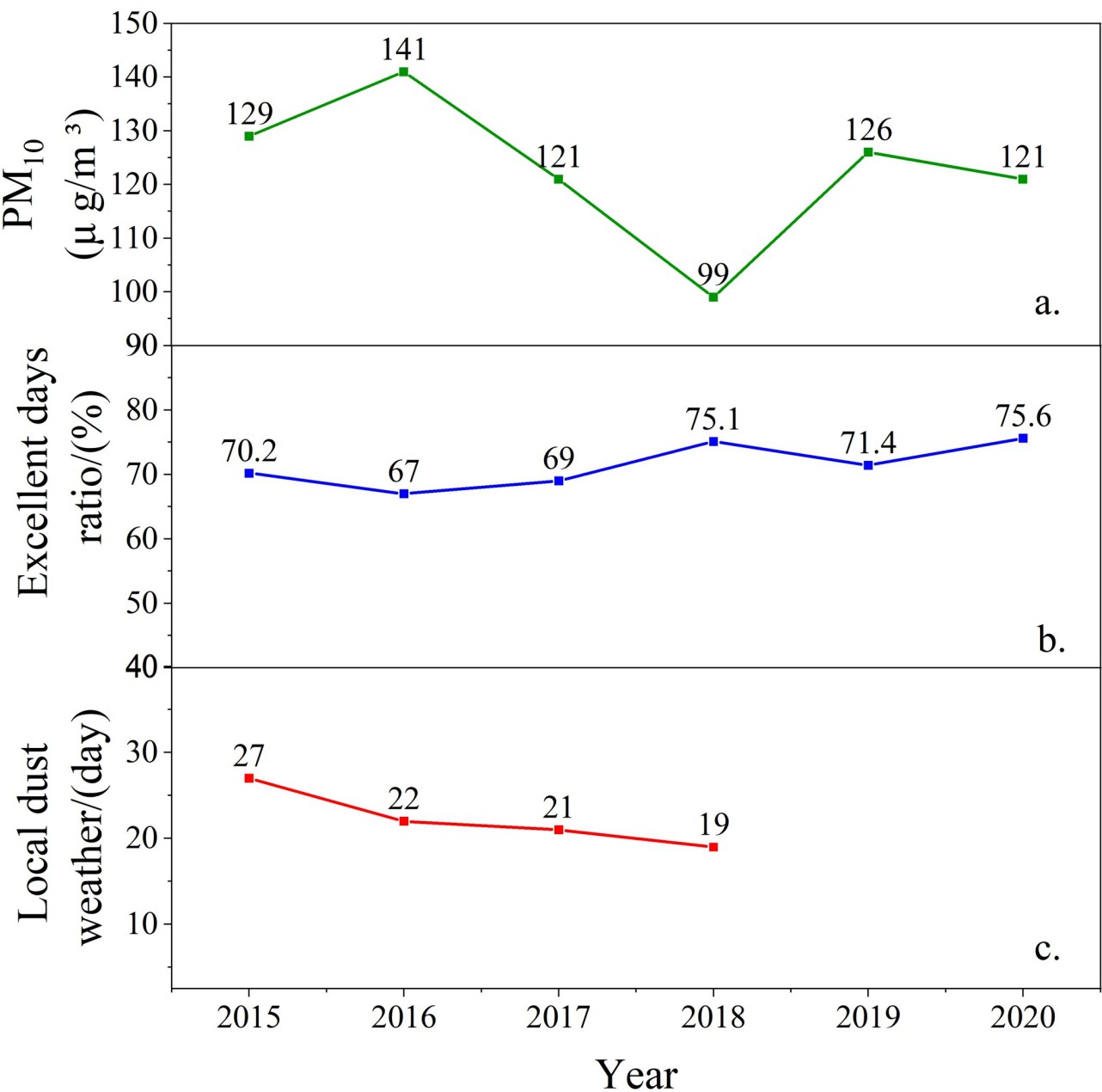

**Fig 9. Statistical chart of sandstorm frequency and air quality from 2015 to 2020.**

dust storms and large natural dust emissions in the Tarim Basin of southern Xinjiang are the main reasons for the high PM$_{2.5}$ values in these areas [52]. However, because the basin is surrounded by mountains in the north, south, and west, the pollutants carried by the air flow are blocked by high mountains when reaching the basin edge and are not easy to spread. Therefore, the whole Tarim Basin and Korla, Aksu, Artux, Kashgar, and Hotan regions on the edge of the Tarim Basin have high PM$_{2.5}$ levels.

Seasonal variations in the PM$_{2.5}$ concentration in Xinjiang are significant, with an average PM$_{2.5}$ concentration of 52.88μg/m$^3$, 38.63μg/m$^3$, 40.51μg/m$^3$, and 66.15μg/m$^3$ for each season,

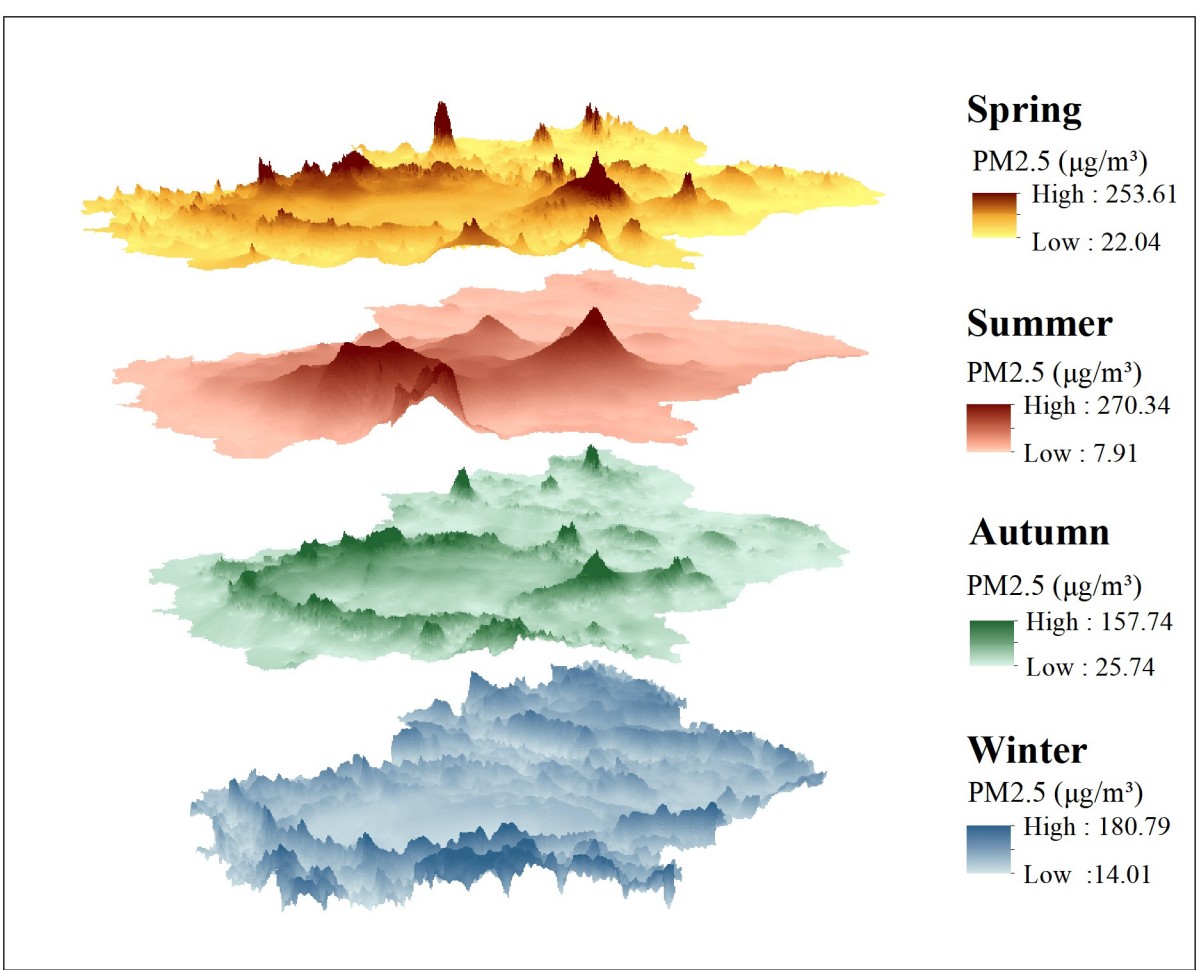

**Fig 10. Spatial distribution map of PM₂.₅ concentration in four seasons inverted by the GTWR model in Xinjiang.**

showing the characteristics of winter>spring>autumn>summer, which is compatible with the research results of Ren [53]. The mean value of PM$_{2.5}$ concentration is the maximum in winter, with the range of 14.01μg/m$^3$-180.79μg/m$^3$, the quarterly average value exceeds the limit of the national secondary standard by 89%. The discharge of large amount of dye particles from coal-burning during winter heating period is one of the causes for the high PM$_{2.5}$ concentration in winter in Xinjiang [54]. The unique topographic features and meteorological conditions of Urumqi cause the pollutants released from coal-burning during winter heating period to diffuse poorly and aggravate the accumulation of pollutants [55]. Therefore, the average seasonal maximum value in 2015–2020 in Xinjiang is during winter, and the majority of high-value distribution locations are in Urumqi and its surroundings.

In summary, this study inverse-performed the PM$_{2.5}$ concentration dataset with 1 km spatial resolution in Xinjiang for 2015–2020 based on the MCD19A2 aerosol product. Compared with the MOD04_3k data with 3 km spatial resolution [56], the MOD04_L2 data with 10 km spatial resolution [57] and the fused data of both commonly used in previous studies, the MCD19A2 AOD data with 1 km resolution used in this paper is improved in resolution and can maintain detailed information on the spatial distribution of AOD. Furthermore, when the same model is used, the accuracy of PM$_{2.5}$ inversion using MCD19A2 data is higher. For

example, Chu et al [58] inverted Taiwan PM$_{2.5}$ data using the GTWR model and MOD04_3k AOD data with an $R^2$ of 0.66, and Shi et al [59] inverted Hong Kong PM$_{2.5}$ data by GTWR model and MOD04_3K and MYD04_3K aerosol data with an $R^2$ of 0.654 Therefore, the MCD19A2 aerosol product with higher data coverage and accuracy can effectively avoid the estimation error in the PM$_{2.5}$ inversion process and is more conducive to the generation of high-resolution PM$_{2.5}$ concentrations.

## Analysis of influencing factors

From the average correlation among meteorological, NDVI, and DEM factors and PM$_{2.5}$ from 2015 to 2020, RH, WS, and DEM have a favorable influence on PM$_{2.5}$ concentration in Xinjiang, whereas PRE, TEM, BLH, NDVI, and ET have a detrimental effect on PM$_{2.5}$ concentration (Fig 11). The strong wind weather is frequent in Xinjiang. A large number of sand dust aerosols diffuse with the wind, thereby increasing the PM$_{2.5}$ levels in the air. When the wind speed reaches a certain strength, loose soil surface forms dust under the strong wind, thereby aggravating the air pollution [60]. As the relative humidity rises, the particles absorb more

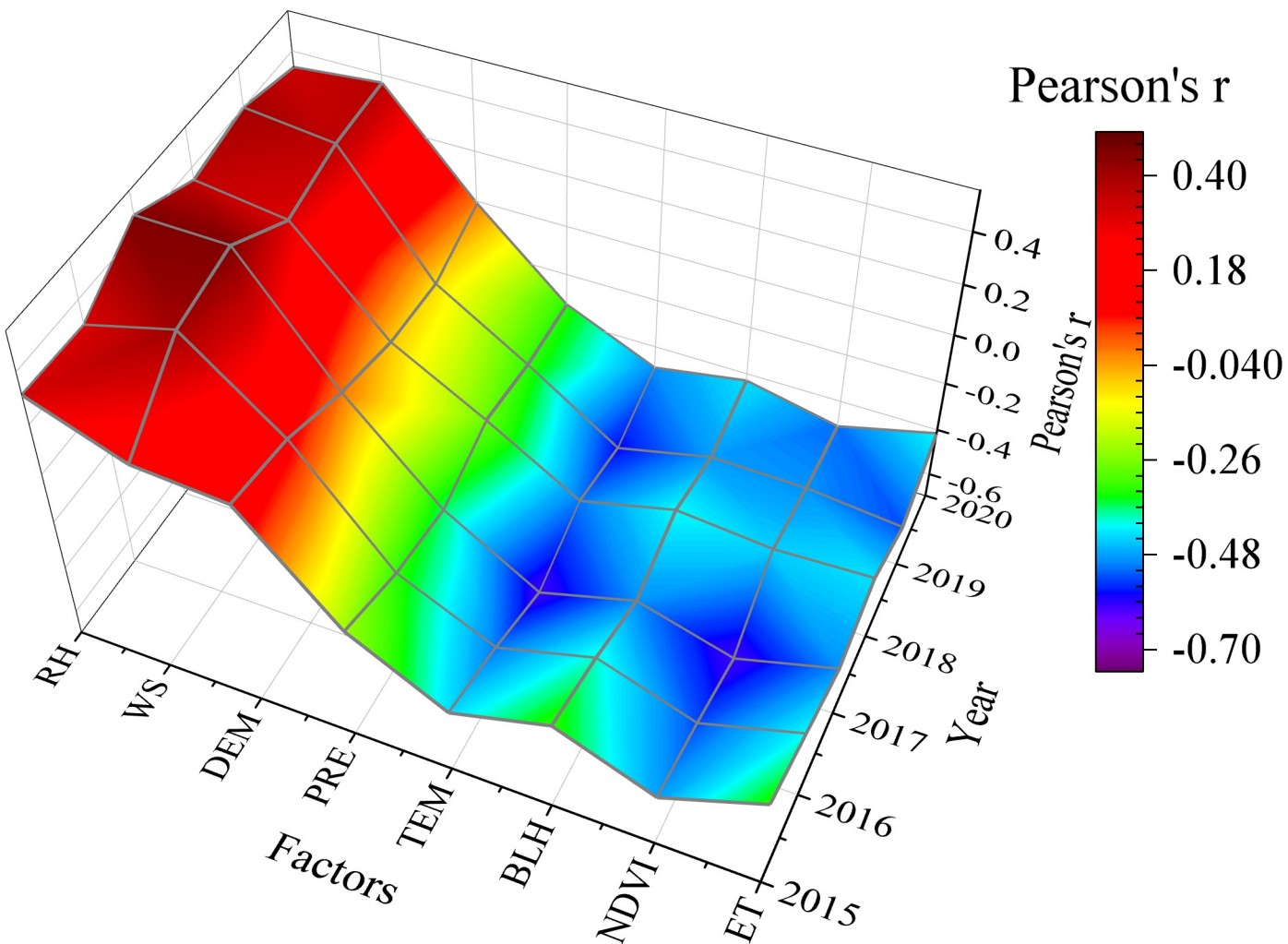

**Fig 11. Correlation map of PM$_{2.5}$ and influencing factors from 2015 to 2020.**

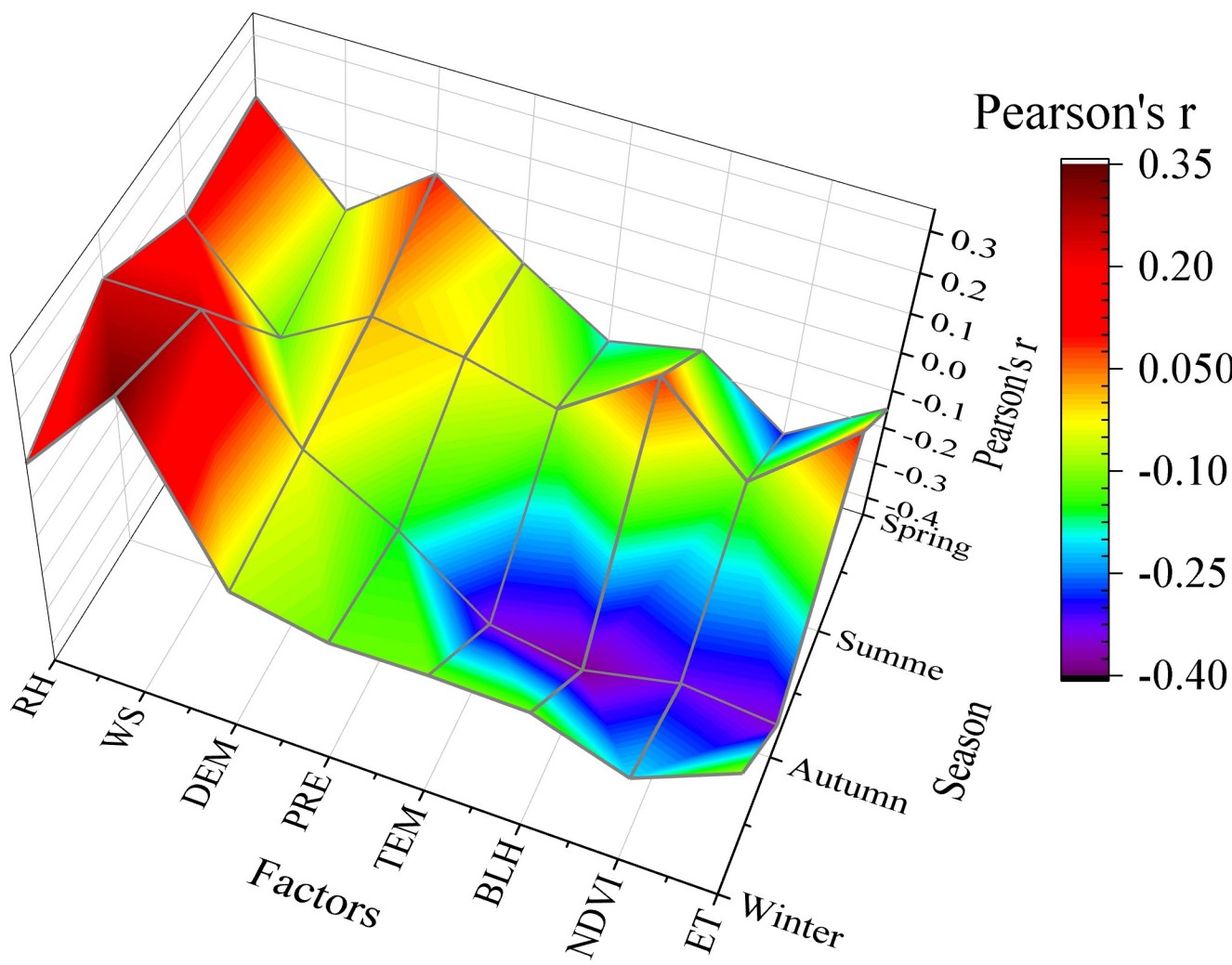

**Fig 12. Correlation map of seasonal PM$_{2.5}$ and influencing factors.**

moisture, which promoted the increase in aerosol content. The rise of temperature and the increase in precipitation promote the atmospheric convection activity and the sedimentation of particles in the air, which can lower the air's content of PM$_{2.5}$ [61]. Furthermore, vegetation can purify the air and play a certain role in reducing PM$_{2.5}$ pollution.

The primary meteorological variables that impact PM$_{2.5}$ also vary in each season (Fig 12). PM$_{2.5}$ in spring is mainly affected by RH, TEM, and NDVI, and their correlations are 0.15, −0.19, and −0.30 respectively. In summer, RH, WS, and NDVI are the main meteorological factors affecting PM$_{2.5}$. RH promotes PM$_{2.5}$, whereas wind can accelerate the diffusion of particulate matter. Thus, it inhibits PM$_{2.5}$. In autumn, except RH and WS, BLH, ET, TEM, and NDVI all inhibit PM$_{2.5}$. Therefore, in the autumn, when the climate is dry, the atmospheric system is stable, and the vegetation is lush, the PM$_{2.5}$ concentration in Xinjiang is low. WS and NDVI are the dominant factors that affect the PM$_{2.5}$ concentration in winter in Xinjiang. The surface layer of soil is lifted by the wind, which raises the particles concentration in the air. However, the temperature in winter in Xinjiang is low, and forming an inversion layer near the ground is easy because of the influence of cold air flow to inhibit the diffusion of pollutants.

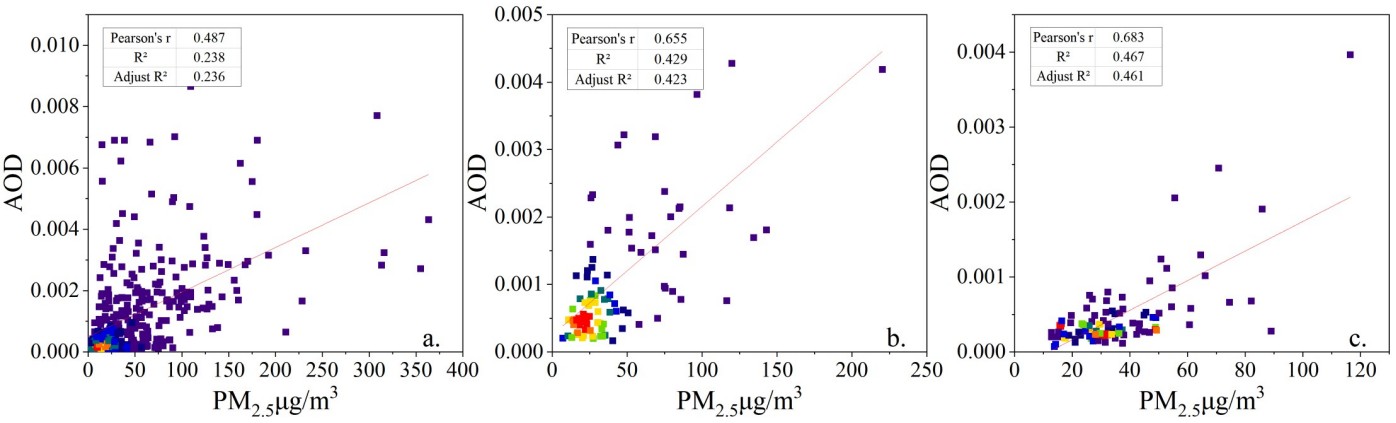

**Fig 13. AOD-PM$_{2.5}$ scatter plot.**

In addition, the sparse vegetation in Xinjiang during winter cannot block particles and purify the air. Therefore, the PM$_{2.5}$ in winter in Xinjiang is the highest among the four seasons. In addition, the stable atmosphere in winter [62] will accelerate the adsorption of aerosols [63], thereby increasing the concentration of pollutants in the air.

## Data uncertainty analysis

In this paper, four cities, Urumqi, Shihezi, Changji and Wujiaqu, were selected as the validation area for March-November 2015, which have a high level of urbanization and more detailed ground observation data, and contain 11 stations in total. Three scenarios were used for uncertainty analysis: scenario I, plotting the scatter plots of daily AOD and daily ground observation PM$_{2.5}$ averages on days 1, 15 and 30 (28, 29 or 31) of each month from March to November 2015 (Fig 13a, cumulative processed image 216 views), with a correlation of 0.487 and an adjusted R$^2$ of 0.236.

Option II, the scatter plot of monthly mean AOD and monthly mean PM$_{2.5}$ from ground observation (Fig 13b, cumulative processed image 216 views), whose correlation is 0.655 and adjusted R$^2$ is 0.423, is plotted by month.

For Scheme III, the scatter plot of the monthly average value of AOD calculated from the daily average value of AOD for March-November and the monthly average value of PM$_{2.5}$ observed on the ground (Fig 13c, cumulative processed image 2184 views) was plotted, and its correlation was 0.683 with an adjusted R$^2$ of 0.461.

The above results show that the correlation between the daily average and the observed daily average is low for Option I, the accuracy of Option III is the highest but the workload is huge, while Option II can both obtain high accuracy and save about 90% of the workload. Therefore, Option II was selected as the data processing option for this study.

## Conclusion

In this study, the spatial distribution of PM$_{2.5}$ concentration in Xinjiang from 2015 to 2020 was estimated by GTWR model, and the following conclusions were drawn:

First, the GTWR model can invert the PM$_{2.5}$ concentration data in Xinjiang more accurately than the GWR and SLR models. Second, MCD19A2 shows the advantage of high spatial resolution in the inversion and spatiotemporal distribution expression. Third, the annual spatial distribution of PM$_{2.5}$ in Xinjiang from 2015 to 2020 is more consistent with the

topographic variations, with high PM$_{2.5}$ concentrations at low elevations and low PM$_{2.5}$ concentrations at high elevations, and the seasonal average concentrations show the characteristics of winter > spring > autumn > summer. Fourth, both official statistics and our inversion results indicate a significant decreasing trend in the annual variation of PM$_{2.5}$ in Xinjiang, which may be related to local environmental protection policies.

## Supporting information

**S1 File.**
(DOCX)

## Acknowledgments

We thank the sponsor for its good role in data collection and analysis and manuscript preparation. Acknowledgments for the data support from "National Earth System Science Data Center, National Science & Technology Infrastructure of China. (http://www.geodata.cn)".

## Author Contributions

**Conceptualization:** Weilin Quan.

**Data curation:** Weilin Quan, Yitu Guo, Wenyue Hai, Jimi Song, Bowen Zhang.

**Formal analysis:** Weilin Quan.

**Funding acquisition:** Nan Xia.

**Methodology:** Weilin Quan.

**Project administration:** Nan Xia.

**Resources:** Nan Xia.

**Software:** Weilin Quan.

**Supervision:** Nan Xia.

**Validation:** Weilin Quan.

**Visualization:** Weilin Quan.

**Writing – original draft:** Weilin Quan.

**Writing – review & editing:** Nan Xia.

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
