## [Decision Letter · Decision Letter 0]

21 Nov 2022

PONE-D-22-28837PM2.5 concentration assessment based on geographical and temporal weighted regression model and MCD19A2 from 2015 to 2020 in Xinjiang, ChinaPLOS ONE

Dear Dr. Xia

Thank you for submitting your manuscript to PLOS ONE. After careful consideration, we feel that it has merit but does not fully meet PLOS ONE’s publication criteria as it currently stands. Therefore, we invite you to submit a revised version of the manuscript that addresses the points raised during the review process.

We look forward to receiving your revised manuscript.

Kind regards,

Chun Liu

Academic Editor

PLOS ONE

3. We note that Figures 1,2,5 and 6 in your submission contain [map/satellite] images which may be copyrighted. All PLOS content is published under the Creative Commons Attribution License (CC BY 4.0), which means that the manuscript, images, and Supporting Information files will be freely available online, and any third party is permitted to access, download, copy, distribute, and use these materials in any way, even commercially, with proper attribution. For these reasons, we cannot publish previously copyrighted maps or satellite images created using proprietary data, such as Google software (Google Maps, Street View, and Earth). For more information, see our copyright guidelines: http://journals.plos.org/plosone/s/licenses-and-copyright.

a. You may seek permission from the original copyright holder of Figures 1,2,5 and 6  to publish the content specifically under the CC BY 4.0 license.   

Reviewers' comments:

Reviewer's Responses to Questions

**Comments to the Author**

1. Is the manuscript technically sound, and do the data support the conclusions?

Reviewer #1: Yes

Reviewer #2: Yes

Reviewer #3: Yes

Reviewer #4: Partly

2. Has the statistical analysis been performed appropriately and rigorously? 

Reviewer #1: Yes

Reviewer #2: Yes

Reviewer #3: Yes

Reviewer #4: No

3. Have the authors made all data underlying the findings in their manuscript fully available?

Reviewer #1: Yes

Reviewer #2: Yes

Reviewer #3: Yes

Reviewer #4: No

4. Is the manuscript presented in an intelligible fashion and written in standard English?

Reviewer #1: Yes

Reviewer #2: Yes

Reviewer #3: Yes

Reviewer #4: No

5. Review Comments to the Author

Reviewer #1: “PM2.5 concentration assessment based on geographical and temporal weighted regression model and MCD19A2 from 2015 to 2020 in Xinjinag, China”

High quality and spatial resolution of PM2.5 data is necessary for assessing the air quality and public health. This study proposes an interesting method for deriving the PM2.5 datasets with high quality and spatial resolution. Considering several issues, I recommend for potential publication after Minor revision.

Specific comments:

Title: the authors should clearly state at what temporal scale, which would be better for readers.

Introduction: Line 75-85, the review of PM2.5 retrieval models is insufficient. Though GTWR is a promising method for PM2.5 estimations, recently, some machine learning and deep learning models can also consider spatiotemporal correlation as well as have a higher estimation accuracy. Therefore, the author need to do a further review of PM2.5 retrieval models and talk more about why the GTWR model was chosen for PM2.5 retrievals. For instance, Predicting annual PM 2.5 in mainland China from 2014 to 2020 using multi temporal satellite product: An improved deep learning approach with spatial generalization ability, ISPRS Journal of Photogrammetry and Remote Sensing.

Method: The authors should give a flowchart on more clearly describing the main method. One issue makes me confusing that the GWR is proposed for considering land surface characteristics, while PM2.5 is a non-normal distribution meteorological variable. So the suitability of GWR family for predicting PM 2.5 needs to be further discussed. While for downscaling satellite-based precipitation products, land surface characteristics were generally utilized using GWR method, for instance, A spatial data mining algorithm for downscaling TMPA 3B43 V7 data over the Qinghai–Tibet Plateau with the effects of systematic anomalies removed, Remote Sensing of Environment.

Figures: The quality of all the Figures should be further improved for meeting the publication standard.

Reviewer #2: I'm finding many problems with the grammar and organization that would lead me to suggest that it still needs editing by someone with some subject area experience. While the topic is good and the analysis is well-done, the importance of the research needs to be better explained.

Section Methods. It makes no sense to describe in detail each of the methods. I recommend that the authors omit all the formulas and describe each of their methods more superficially.

The article lacks a critical analysis of the change in PM2.5 and AOD. Pay more attention to the actual variation of these pollutants in different areas of the province. It would be helpful if the authors focused more on dust aerosols and the impact of dust storms. Enough papers have already been published on this topic, so the authors can cite one of published articles.

Why don't the authors mention the accuracy of MODIS products. Moreover, there are many studies on the mutual verification of MODIS and AERONET (Bilal et al., 2019; Filonchyk and Hurynovich, 2020).

"Fig 2. AOD interpolation diagram from 2015-2020". Looking at Figure 2, I noticed that it has nothing to do with the diagram. As I see it, this is the spatial distribution of the AOD.

Why don't authors use the same color scheme for all figures?

References：

Filonchyk, M., & Hurynovich, V. (2020). Validation of MODIS aerosol products with AERONET measurements of different land cover types in areas over Eastern Europe and China. Journal of Geovisualization and Spatial Analysis, 4(1), 1-11.

Bilal, M., Nazeer, M., Nichol, J., Qiu, Z., Wang, L., Bleiweiss, M. P., ... & Lolli, S. (2019). Evaluation of Terra-MODIS C6 and C6. 1 aerosol products against Beijing, XiangHe, and Xinglong AERONET sites in China during 2004-2014. Remote Sensing, 11(5), 486.

Reviewer #3: This manuscript compares several model’s performance in retrieving PM2.5,and give a spatial and temporal patten of PM2.5, it would be better if the research content go a little deeper. Some problems need to be solved:

1. In line 171, you did’t give the mathematical notation for the humidity influence factor.

2. In line 186, the symbol for the random variable disappeared.

3. Table 3 needs to be adjusted to be more aesthetically pleasing.

4. In line 248, the comma was in a wrong place.

5. Does Figure 4 use multi-year average data for analysis? It is not clearly marked.

6. The sentence in line 315, it’s unclear for understanding.

7. The discussion needs to be improved.

Reviewer #4: Review on the manuscript titled “PM2.5 concentration assessment based on geographical and temporal weighted regression model and MCD19A2 from 2015 to 2020 in Xin-jiang, China”.

In this study, the authors applied the SLR, GWR, and GTWR models with MCD19A2, NDVI, DEM, and meteorological data to retrieve PM2.5 concentrations from year 2015 to 2020 in Xinjiang. The performance of each model is compared, the optimal model (GTWR) is selected for PM2.5 inversion. The spatiotemporal distribution characteristics of PM2.5 and meteorological impact factors are further analyzed. This topic is interesting; however, the manuscript is not well organized and the writing of the paper is not good, which makes the manuscript looking more like a project report instead of a scientific research paper. I thinks the paper needs a lot of work to be improved.

Specific comments:

1. All abbreviations appearing at the first time (including abstract) should be given full expression.

2. What is the definition of “small- and medium-size area” ? Is Xijiang (with area of 166×104 km2) a small- and medium-size area? I cannot agree with this subjective description.

3. Line 93-Line107. This paragraph looks like a translation from a project report, which is irrelevant to the major research aim of this paper. Readers cannot get any implications from this paragraph why Xijiang is selected.

4. Line 109: It is better to change “Materials and Methods” to “Data and Method”.

5. Line 114-115 “After excluding the sites with many missing monitoring data and many invalid values”, this description seems very casual. How many sites are in the original dataset? What principle is used to conduct the data quality control? How many sites are deleted?

6. Line 116-118, the authors mentioned that “To match the AOD data in time and space, the average PM2.5 value of each station one hour before and after the satellite transit is calculated to determine the daily, monthly, quarterly, and annual averages of each station”. Using 1-2 hours data to determine the daily, monthly, quarterly and annual averages will raise large uncertainties.

7. Why do the authors only download AOD data for 1st, 15th and 30th of each month?

8. Line 171, which parameter represents the humidity influence factor? “f(RH)” is missing.

9. Line 186, which is a random variable?

10. “Results” and “Discussion” can be combines to Section “Results and Discussions”.

11. What is “AICc”? there is no definition at all.

12. Section “PM2.5 Year Distribution in Xinjing”, this section uses lots of words to describe the phenomenon, without any further explanations.

13. Suggest to move “discussions” to “results” and combine to one Section “Results and Discussions”

14. Conclusion: the 3 paragraphs seems strange, please combine the three paragraphs and conclude the major findings of this study.

15. There are quite a lot of grammar errors or typos, the language should be polished thoroughly.

6. PLOS authors have the option to publish the peer review history of their article (what does this mean?). If published, this will include your full peer review and any attached files.

Reviewer #1: No

Reviewer #2: No

Reviewer #3: No

Reviewer #4: No

---

## [Author Response · Author response to Decision Letter 0]

15 Jan 2023

Responses to reviewer’s comments

NOTE: RC=Reviewer’s comment; AR=Authors’ response

Reviewer(s) Comments:

Reviewer #1

High quality and spatial resolution of PM2.5 data is necessary for assessing the air quality and public health. This study proposes an interesting method for deriving the PM2.5 datasets with high quality and spatial resolution. Considering several issues, I recommend for potential publication after Minor revision.

AR:Thank you very much for your kindly comments on our manuscript. There is no doubt that these comments are valuable and very helpful for revising and improving our manuscript. In what follows, we would like to answer the questions you mentioned and give detailed account of the changes made to the original manuscript.

RC:[1]Title: the authors should clearly state at what temporal scale, which would be better for readers.

AR:The title of this paper has been modified to:PM2.5 concentration assessment based on geographical and temporal weighted regression model and MCD19A2 from 2015 to 2020 in Xinjiang, China.We are very much appreciated for your constructive comments and insightful suggestion.

RC:[2]Introduction: Line 75-85, the review of PM2.5 retrieval models is insufficient. Though GTWR is a promising method for PM2.5 estimations, recently, some machine learning and deep learning models can also consider spatiotemporal correlation as well as have a higher estimation accuracy. Therefore, the author need to do a further review of PM2.5 retrieval models and talk more about why the GTWR model was chosen for PM2.5 retrievals. For instance, Predicting annual PM 2.5 in mainland China from 2014 to 2020 using multi temporal satellite product: An improved deep learning approach with spatial generalization ability, ISPRS Journal of Photogrammetry and Remote Sensing.

AR:Thank you for the helpful suggestion.We have further reviewed the model of the PM2.5 inversion, and discussed the reasons for choosing GTWR model in "Introduction".

RC:[3]Method: The authors should give a flowchart on more clearly describing the main method. One issue makes me confusing that the GWR is proposed for considering land surface characteristics, while PM2.5 is a non-normal distribution meteorological variable. So the suitability of GWR family for predicting PM2.5 needs to be further discussed. While for downscaling satellite-based precipitation products, land surface characteristics were generally utilized using GWR method, for instance, A spatial data mining algorithm for downscaling TMPA 3B43 V7 data over the Qinghai–Tibet Plateau with the effects of systematic anomalies removed, Remote Sensing of Environment.

AR:This is very helpful and insightful suggestion. Thank you so much.We have drawn a flow chart to more clearly describe the main methods (Figure 3).We think about the applicability of GWR prediction in PM2.5: First, in our experiments, GWR has good inversion accuracy (six year average R2=0.64, RMSE=23.71); Secondly, predecessors used GWR model to retrieve PM2.5 concentrations in some regions of China and obtained good results, such as Rui, etc. We have added a reference to this paper in the paper; Finally, we understand the distribution of PM2.5 in this way. Xinjiang covers a large area but has a small population. As a whole, PM2.5 is more affected by natural factors. When natural factors are the dominant factors, PM2.5 presents a normal distribution, while in some local areas, such as urban agglomerations on the northern slope of Tianshan Mountains, PM2.5 is mainly affected by human factors and presents a non normal distribution. Therefore, whether the existence of PM2.5 in Xinjiang is normal or non normal, It is worth further discussion and research. This is only our speculation based on our understanding of the overview of the study area, and it is only for the reviewers' criticism. Based on the above reasons, we chose the GWR model.

Figure 3. Workflow outline

RC:[4]Figures: The quality of all the Figures should be further improved for meeting the publication standard.

AR:Thank you very much!Thank you.We have increased the resolution of each figure in the paper to 300dpi, and tested the quality of each figure in PACE. There is no ambiguity of the figure. However, after uploading figures and generating PDF files in the submission system, the resolution of each figures in the PDF file will be reduced. Please click the picture download link in the upper right corner of the PDF file to obtain high-resolution and high-quality figures.

NOTE: RC = Reviewer’s comment; AR = Authors’ response

Reviewer(s) Comments:

Reviewer #2

I'm finding many problems with the grammar and organization that would lead me to suggest that it still needs editing by someone with some subject area experience. While the topic is good and the analysis is well-done, the importance of the research needs to be better explained.

AR:Thank you very much for your useful comments on our manuscript .We are very sorry for the mistakes in this manuscript and inconvenience they caused in your reading. The manuscript has been thoroughly revised and edited by a native speaker, so we hope it can meet the journal’s standard. 

RC:[1]Section Methods. It makes no sense to describe in detail each of the methods. I recommend that the authors omit all the formulas and describe each of their methods more superficially.

AR:Thank you very much for your valuable suggestions! According to your suggestion, we have briefly described each method, but we think that the formula is necessary, so we have omitted the formula as much as possible, and only the main part has been retained.

RC:[2]The article lacks a critical analysis of the change in PM2.5 and AOD. Pay more attention to the actual variation of these pollutants in different areas of the province. It would be helpful if the authors focused more on dust aerosols and the impact of dust storms. Enough papers have already been published on this topic, so the authors can cite one of published articles.

AR:Thank you for the helpful suggestion.Based on your suggestions, we have supplemented the critical analysis of changes in PM2.5 and AOD[now in lines 281-288], and cited the following papers at appropriate locations:

[1].Liu, J., Ding, J., Rexiding, M., Li, X., Zhang, J., Ran, S., Bao, Q., & Ge, X. (2021). Characteristics of dust aerosols and identification of dust sources in Xinjiang, China. Atmospheric Environment, 262, 118651. [now in lines 349]

RC:[3]Why don't the authors mention the accuracy of MODIS products. Moreover, there are many studies on the mutual verification of MODIS and AERONET (Bilal et al., 2019; Filonchyk and Hurynovich, 2020).

AR:Thank you very much! According to your suggestion, we have supplemented the description of data accuracy in "Data and Method" and cited the following papers:

[1].Filonchyk, M., & Hurynovich, V. (2020). Validation of MODIS aerosol products with AERONET measurements of different land cover types in areas over Eastern Europe and China. Journal of Geovisualization and Spatial Analysis, 4(1), 1-11.

[2].Bilal, M., Nazeer, M., Nichol, J., Qiu, Z., Wang, L., Bleiweiss, M. P., ... & Lolli, S. (2019). Evaluation of Terra-MODIS C6 and C6. 1 aerosol products against Beijing, XiangHe, and Xinglong AERONET sites in China during 2004-2014. Remote Sensing, 11(5), 486. [now in lines 125]

RC:[4]"Fig 2. AOD interpolation diagram from 2015-2020". Looking at Figure 2, I noticed that it has nothing to do with the diagram. As I see it, this is the spatial distribution of the AOD.

AR:Thank you. We have changed the name of the figure from"Fig 2. AOD interpolation diagram from 2015-2020" to "Fig 2. Spatial distribution map of AOD in Xinjiang from 2015-2020"

RC:[5]Why don't authors use the same color scheme for all figures?

AR:Thank you.Since Figure 2 represents the distribution of AOD, and Figure 5 and Figure 6(now Figure 6 and Figure 7) represent the distribution of PM2.5, their attributes are different, so we use different colors to distinguish them. For Figure 9(now Fogure 10), if the same color is used, the boundary of the area in Figure 9 will be blurry and difficult to distinguish, because the same color band represents the same value range, while the maximum and minimum values of PM2.5 in four seasons are different, and the same color represents different value ranges, which will lead to no comparability. For example, the maximum value in summer is 270.34μg/m3, the maximum value in autumn is 157.74μg/m3, when using the same color band, the values represented by the same color are different. Therefore, in consideration of the aesthetics of the drawing and the distinguishability of the pictures, we chose different colors for drawing.

NOTE: RC=Reviewer’s comment; AR=Authors’ response

Reviewer(s) Comments:

Reviewer #3

This manuscript compares several model’s performance in retrieving PM2.5,and give a spatial and temporal patten of PM2.5, it would be better if the research content go a little deeper. Some problems need to be solved:

AR:We feel great thanks for your professional review work on our article. According to your nice suggestions, we have made extensive corrections to our previous draft, the detailed corrections are listed below:

RC:[1] In line 171, you did’t give the mathematical notation for the humidity influence factor.

AR:Thank you for pointing this out. We added the symbol of humidity influencing factor in line 171.[now in lines 165]

RC:[2] In line 186, the symbol for the random variable disappeared.

AR:Thank you for pointing this out.We added the symbol of the random variable in line 186.[now in lines 175]

RC:[3] Table 3 needs to be adjusted to be more aesthetically pleasing.

AR:Thank you for the helpful suggestion.We have adjusted Table 3.

RC:[4] In line 248, the comma was in a wrong place.

AR:Thank you for pointing this out.We have corrected the use of 248 lines of commas.[now in lines 219]

RC:[5] Does Figure 4 use multi-year average data for analysis? It is not clearly marked.

AR:Thank you.Figure 4 (now Figure 5) is analyzed using the six-year average data of each season. According to your suggestion, we have added a description of the data used in Figure 4.[now in lines 244-245]

RC:[6] The sentence in line 315, it’s unclear for understanding.

AR:Thank you very much！We adjusted the 315 line statement to make it easier to understand.[now in lines 304-305]

RC:[7] The discussion needs to be improved.

AR:Thanks for your valuable counsel.The discussion section has been improved and incorporated into "Results and Discussion"

NOTE: RC=Reviewer’s comment; AR=Authors’ response

Reviewer(s) Comments:

Reviewer #4

In this study, the authors applied the SLR, GWR, and GTWR models with MCD19A2, NDVI, DEM, and meteorological data to retrieve PM2.5 concentrations from year 2015 to 2020 in Xinjiang. The performance of each model is compared, the optimal model (GTWR) is selected for PM2.5 inversion. The spatiotemporal distribution characteristics of PM2.5 and meteorological impact factors are further analyzed. This topic is interesting; however, the manuscript is not well organized and the writing of the paper is not good, which makes the manuscript looking more like a project report instead of a scientific research paper. I thinks the paper needs a lot of work to be improved.

AR:Thank you very much for your kindly comments on our manuscript. These comments are all valuable and very helpful for revising and improving our paper, as well as the important guiding significance to our researches. We have studied comments carefully and have made corrections：

RC:[1] All abbreviations appearing at the first time (including abstract) should be given full expression.

AR:Thank you for pointing this out.We have supplemented the full description of all abbreviations appearing for the first time in the text.include:

(1)MCD19A2 (MODIS/Terra+Aqua Land Aerosol Optical Thickness Daily L2G Global 1km SIN Grid V006) 

(2)MODIS (Moderate-resolution Imaging Spectroradiometer)

(3)MOD04_L2 (MODIS Terra/Aqua Aerosol 5-Min L2 Swath 10km)

(4)MOD04_3K (MODIS Terra/Aqua Aerosol 5-Min L2 Swath 3km)

(5)MAIAC (Multi-Angle Implementation of Atmospheric Correction)

(6)National Aeronautics and Space Administration (NASA) 

(7)European Centre for Medium-Range Weather Forecasts (ECMWF)

(8)Akaike information criterion（AICc)

RC:[2]What is the definition of “small- and medium-size area” ? Is Xijiang (with area of 166×104 km2) a small- and medium-size area? I cannot agree with this subjective description.

AR:Thank you for your comments. We corrected the subjective description of "small and medium-sized areas" in the text.

RC:[3]Line 93-Line107. This paragraph looks like a translation from a project report, which is irrelevant to the major research aim of this paper. Readers cannot get any implications from this paragraph why Xijiang is selected.

AR:Thank you very much for your constructive comments and suggestions! We have revised lines 93-107, focusing on the reasons for choosing Xinjiang. 

[now in lines 91-107]

RC:[4]Line 109: It is better to change “Materials and Methods” to “Data and Method”.

AR:The "Materials and Methods" has been revised to “Data and Method”.Thank you again.[now in lines 109]

RC:[5] Line 114-115 “After excluding the sites with many missing monitoring data and many invalid values”, this description seems very casual. How many sites are in the original dataset? What principle is used to conduct the data quality control? How many sites are deleted?

AR:Thank you for pointing this out.Our original data includes 35 stations in total. Since the city environmental monitoring station in Kashgar region (in the southwest of Xinjiang) has four months of missing PM2.5 data, and the other monitoring station WuBan , which has 73 months of missing or invalid data, we deleted these two stations. In the data quality control, we adopt the principle that the PM2.5 value remains unchanged for more than 12 consecutive hours, the hourly concentration value deviates by more than 3 times the standard deviation of the 24 hour mean value of the mass concentration of the day, and the mass concentration value is greater than 999μg/m3 as invalid values. Cite line xx in the form of reference.[now in lines 113-116]

RC:[6]Line 116-118, the authors mentioned that “To match the AOD data in time and space, the average PM2.5 value of each station one hour before and after the satellite transit is calculated to determine the daily, monthly, quarterly, and annual averages of each station”. Using 1-2 hours data to determine the daily, monthly, quarterly and annual averages will raise large uncertainties.

AR:Thanks for your valuable counsel.There is only one MCD19A2 data every day. We need to match the ground monitoring data according to the satellite transit time. This uncertainty cannot be avoided at present, so we have minimized this uncertainty.

RC:[7]Why do the authors only download AOD data for 1st, 15th and 30th of each month?

AR:Thank you for your valuable comments. Using the mean value of AOD data at the beginning, middle and end of a month to replace the monthly mean value will lead to some uncertainties. Ideally, we should use daily data for the experiment, but the amount of data will be very large, which is not conducive to the experiment. In addition, we focus more on the feasibility of model inversion of PM2.5 in our research. Later, we can use the AOD dataset of complete time series to synthesize PM2.5 products, so as to improve the accuracy of data.

RC:[8]Line 171, which parameter represents the humidity influence factor? “f(RH)” is missing.

AR:Thank you for pointing this out.We have added the missing "f (RH)" in line 171.[now in lines 165]

RC:[9]Line 186, which is a random variable?

AR:In line 186, "" represents a random variable. We have added it to this line. Thank you again for your valuable comments.[now in lines 175]

RC:[10]“Results” and “Discussion” can be combines to Section “Results and Discussions”.

AR:Thanks for your valuable counsel.We have moved the discussion to the results section and merged them into "results and discussion".

RC:[11]What is “AICc”? there is no definition at all.

AR:Thank you for pointing this out.We have supplemented the definition of AICc in "Model Comparison".[now in lines 219-221]

RC:[12]Section “PM2.5 Year Distribution in Xinjing”, this section uses lots of words to describe the phenomenon, without any further explanations.

AR:Thank you for the helpful suggestion,We have further explained and discussed this phenomenon in the section "Annual distribution of PM2.5 in Xinjiang", and specifically analyzed PM2.5 with DEM and official data [now in lines 281-288].

RC:[13]Suggest to move “discussions” to “results” and combine to one Section “Results and Discussions”

AR:Thanks for your valuable counsel.We have moved the discussion to the results section and merged them into "results and discussion".

RC:[14]Conclusion: the 3 paragraphs seems strange, please combine the three paragraphs and conclude the major findings of this study.

AR:Thank you for the helpful suggestion. We further summarize the main findings of this study.

RC:[15]There are quite a lot of grammar errors or typos, the language should be polished thoroughly.

AR:Thank you very much for your useful comments on our manuscript .We are very sorry for the mistakes in this manuscript and inconvenience they caused in your reading. The manuscript has been thoroughly revised and edited by a native speaker, so we hope it can meet the journal’s standard.

---

## [Decision Letter · Decision Letter 1]

13 Feb 2023

PONE-D-22-28837R1PM2.5 concentration assessment based on geographical and temporal weighted regression model and MCD19A2 from 2015 to 2020 in Xinjiang, ChinaPLOS ONE

Dear Dr. Xia,

Thank you for submitting your manuscript to PLOS ONE. After careful consideration, we feel that it has merit but does not fully meet PLOS ONE’s publication criteria as it currently stands. Therefore, we invite you to submit a revised version of the manuscript that addresses the points raised during the review process.

We look forward to receiving your revised manuscript.

Kind regards,

Chun Liu

Academic Editor

PLOS ONE

Reviewers' comments:

Reviewer's Responses to Questions

**Comments to the Author**

1. If the authors have adequately addressed your comments raised in a previous round of review and you feel that this manuscript is now acceptable for publication, you may indicate that here to bypass the “Comments to the Author” section, enter your conflict of interest statement in the “Confidential to Editor” section, and submit your "Accept" recommendation.

Reviewer #2: All comments have been addressed

Reviewer #4: (No Response)

2. Is the manuscript technically sound, and do the data support the conclusions?

Reviewer #2: Yes

Reviewer #4: Partly

3. Has the statistical analysis been performed appropriately and rigorously? 

Reviewer #2: Yes

Reviewer #4: Yes

4. Have the authors made all data underlying the findings in their manuscript fully available?

Reviewer #2: Yes

Reviewer #4: No

5. Is the manuscript presented in an intelligible fashion and written in standard English?

Reviewer #2: Yes

Reviewer #4: Yes

6. Review Comments to the Author

Reviewer #2: The authors have done a good job of improving their manuscript. Therefore, I recommend this article and hope that this article will contribute to the scientific community.

Reviewer #4: The authors have carefully addressed the reviewers’ comments and the quality of the manuscript has been substantially improved. However, my major concern is that this study only used AOD data from the first, 15th, and 30th of each month in Xinjiang from 2015 to 2020. In this case, the results describing trends of PM2.5 will raise large uncertainties. At least detailed comparisons between your retrieved monthly/yearly average results at specific grids (covering the surface sites) and the ground observations covering the long time period should be given. The uncertainties should be carefully discussed. In addition, there are some minor typos that need to be revised.

1. Line 110 “Data” can be deleted.

2. Line 298, two “Fig7”.

3. The figures are not clear.

7. PLOS authors have the option to publish the peer review history of their article (what does this mean?). If published, this will include your full peer review and any attached files.

Reviewer #2: No

Reviewer #4: No

---

## [Author Response · Author response to Decision Letter 1]

28 Mar 2023

Responses to reviewer’s comments

NOTE: RC=Reviewer’s comment; AR=Authors’ response

Reviewer(s) Comments:

Reviewer #2

The authors have done a good job of improving their manuscript. Therefore, I recommend this article and hope that this article will contribute to the scientific community.

AR:Thank you very much for your kind work and recognition of our manuscript！Your suggestions have been very helpful in modifying and improving our manuscript.

NOTE: RC = Reviewer’s comment; AR = Authors’ response

Reviewer(s) Comments:

Reviewer #4

The authors have carefully addressed the reviewers’ comments and the quality of the manuscript has been substantially improved. However, my major concern is that this study only used AOD data from the first, 15th, and 30th of each month in Xinjiang from 2015 to 2020. In this case, the results describing trends of PM2.5 will raise large uncertainties. At least detailed comparisons between your retrieved monthly/yearly average results at specific grids (covering the surface sites) and the ground observations covering the long time period should be given. The uncertainties should be carefully discussed. In addition, there are some minor typos that need to be revised.

AR:Thank you very much for your kindly comments on our manuscript. There is no doubt that these comments are valuable and very helpful for revising and improving our manuscript. In what follows, we would like to answer the questions you mentioned and give detailed account of the changes made to the original manuscript.

RC:[1]At least detailed comparisons between your retrieved monthly/yearly average results at specific grids (covering the surface sites) and the ground observations covering the long time period should be given. The uncertainties should be carefully discussed.

AR:Thank you very much for your valuable suggestions.We compared the monthly/yearly average results at specific grids (covering the surface sites) with the ground observation results covering a long period of time , and discussed the uncertainties in the Data uncertainty analysis. [now in lines 402-420]

RC:[2]Line 110 “Data” can be deleted.

AR:Thank you very much! We have deleted "DATA" on line 110.

RC:[3] Line 298, two “Fig7”.

AR:Thank you for pointing this out.We deleted the redundant "Fig 7" on line 298.

RC:[4]The figures are not clear.

AR:Thank you.We have increased the resolution of each figure in the paper to 300-500dpi, and tested the quality of each figure in PACE. There is no ambiguity of the figure. However, after uploading figures and generating PDF files in the submission system, the resolution of each figures in the PDF file will be reduced. Please click the picture download link in the upper right corner of the PDF file to obtain high-resolution and high-quality figures.

---

## [Decision Letter · Decision Letter 2]

27 Apr 2023

PM2.5 concentration assessment based on geographical and temporal weighted regression model and MCD19A2 from 2015 to 2020 in Xinjiang, China

PONE-D-22-28837R2

Dear Dr. Xia,

We’re pleased to inform you that your manuscript has been judged scientifically suitable for publication and will be formally accepted for publication once it meets all outstanding technical requirements.

Kind regards,

Chun Liu

Academic Editor

PLOS ONE

Additional Editor Comments (optional):

Reviewers' comments:

Reviewer's Responses to Questions

**Comments to the Author**

1. If the authors have adequately addressed your comments raised in a previous round of review and you feel that this manuscript is now acceptable for publication, you may indicate that here to bypass the “Comments to the Author” section, enter your conflict of interest statement in the “Confidential to Editor” section, and submit your "Accept" recommendation.

Reviewer #2: (No Response)

2. Is the manuscript technically sound, and do the data support the conclusions?

Reviewer #2: Yes

3. Has the statistical analysis been performed appropriately and rigorously? 

Reviewer #2: Yes

4. Have the authors made all data underlying the findings in their manuscript fully available?

Reviewer #2: No

5. Is the manuscript presented in an intelligible fashion and written in standard English?

Reviewer #2: Yes

6. Review Comments to the Author

Reviewer #2: (No Response)

7. PLOS authors have the option to publish the peer review history of their article (what does this mean?). If published, this will include your full peer review and any attached files.

Reviewer #2: No

---

## [Editor Report · Acceptance letter]

2 May 2023

PONE-D-22-28837R2 

PM_2.5_ concentration assessment based on geographical and temporal weighted regression model and MCD19A2 from 2015 to 2020 in Xinjiang, China 

Dear Dr. Xia:

I'm pleased to inform you that your manuscript has been deemed suitable for publication in PLOS ONE. Congratulations! Your manuscript is now with our production department. 

Kind regards, 

on behalf of

Dr. Chun Liu 

Academic Editor

PLOS ONE